# Cryo-EM structure of the NDH–PSI–LHCI supercomplex from *Spinacia oleracea*

Bianca Introini ®[1], Alexander Hahn[1,2] & Werner Kühlbrandt ®[1] ✉

The nicotinamide adenine dinucleotide phosphate (NADPH) dehydrogenase (NDH) complex is crucial for photosynthetic cyclic electron flow and respiration, transferring electrons from ferredoxin to plastoquinone while transporting $H^+$ across the chloroplast membrane. This process boosts adenosine triphosphate production, regardless of NADPH levels. In flowering plants, NDH forms a supercomplex with photosystem I, enhancing its stability under high light. We report the cryo-electron microscopy structure of the NDH supercomplex in *Spinacia oleracea* at a resolution of 3.0–3.3 Å. The supercomplex consists of 41 protein subunits, 154 chlorophylls and 38 carotenoids. Subunit interactions are reinforced by 46 distinct lipids. The structure of NDH resembles that of mitochondrial complex I closely, including the quinol-binding site and an extensive internal aqueous passage for proton translocation. A well-resolved catalytic plastoquinone (PQ) occupies the PQ channel. The pronounced structural similarity to complex I sheds light on electron transfer and proton translocation within the NDH supercomplex.

Photosynthesis is the vital process through which green plants, algae and cyanobacteria harness solar energy to power cellular metabolism and assimilate carbon dioxide. The initial light reactions in the thylakoid membrane convert solar energy into the chemical energy of a phosphate bond in ATP (adenosine triphosphate) and generate NADPH (nicotinamide adenine dinucleotide phosphate). The subsequent dark reactions of the Calvin cycle in the chloroplast stroma fix $CO_2$ and produce organic compounds. The two processes are coordinated by linear or cyclic electron flow. In linear electron flow, electrons are transported through a sequence of redox-active electron transfer complexes, most notably photosystems (PSs) I and II, generating a proton gradient across the thylakoid membrane and producing NADPH. The proton gradient drives ATP production by ATP synthase. ATP and NADPH are produced at a ratio of approximately 1.3 (refs. 1,2). Cyclic electron flow regulates the ATP:NADPH ratio and protects the photosynthetic machinery from stress-induced damage, cycling electrons around PSI[3,4]. In the thylakoid membranes of plants and cyanobacterial chloroplasts, NADPH dehydrogenase (NDH) is homologous to the mitochondrial NADH:ubiquinone (Q) oxidoreductase (complex I). NDH transfers electrons from PSI to the plastoquinone (PQ) pool

and cytochrome $b_6f$ complex, pumping protons across the thylakoid membrane. Cyclic electron flow enhances ATP production by adding to the proton-motive force.

The 11 subunits of plant NDH are homologous to the core subunits of complex I[5]. The sequences of subunits NdhA–NdhG in the NDH membrane arm resemble the proximal ($P_P$) and distal ($P_D$) modules in the membrane arm of mitochondrial complex I (ND1–ND6). Subunits NdhH–NdhK in the peripheral arm of NDH are homologous to the Q reduction module of complex I (NDFUS2, NDFUS3, NDFUS7 and NDFUS8; Extended Data Fig. 1). Unlike complex I, NDH lacks an NADH oxidation (N) center and uses ferredoxin (Fd) as the electron donor instead. Subunits NdhL–NdhO and NdhS–NdhV of plant NDH are specific to oxygenic photosynthesis. Subunits PnsL1–PnsL5 and PnsB1–PnsB5 are chloroplast specific.

The existence of the NDH–PSI–light-harvesting complex I (LHCI) supercomplex in plants has been demonstrated by blue native PAGE and immunoblotting[6]. Mass spectrometry indicated the subunits that are crucial for supercomplex formation[6]. Lhca5 and Lhca6 have key roles in supercomplex stability. Electron microscopy (EM)[7] and cryo-EM[8,9] revealed that one or two copies of PSI are attached to one NDH complex

[1]Department of Structural Biology, Max Planck Institute of Biophysics, Frankfurt am Main, Germany. [2]Present address: MVZ am Helios Klinikum, Emil von Behring GmbH, Institut für Gewebediagnostik/Pathologie, Berlin, Germany. ✉e-mail: werner.kuehlbrandt@biophys.mpg.de

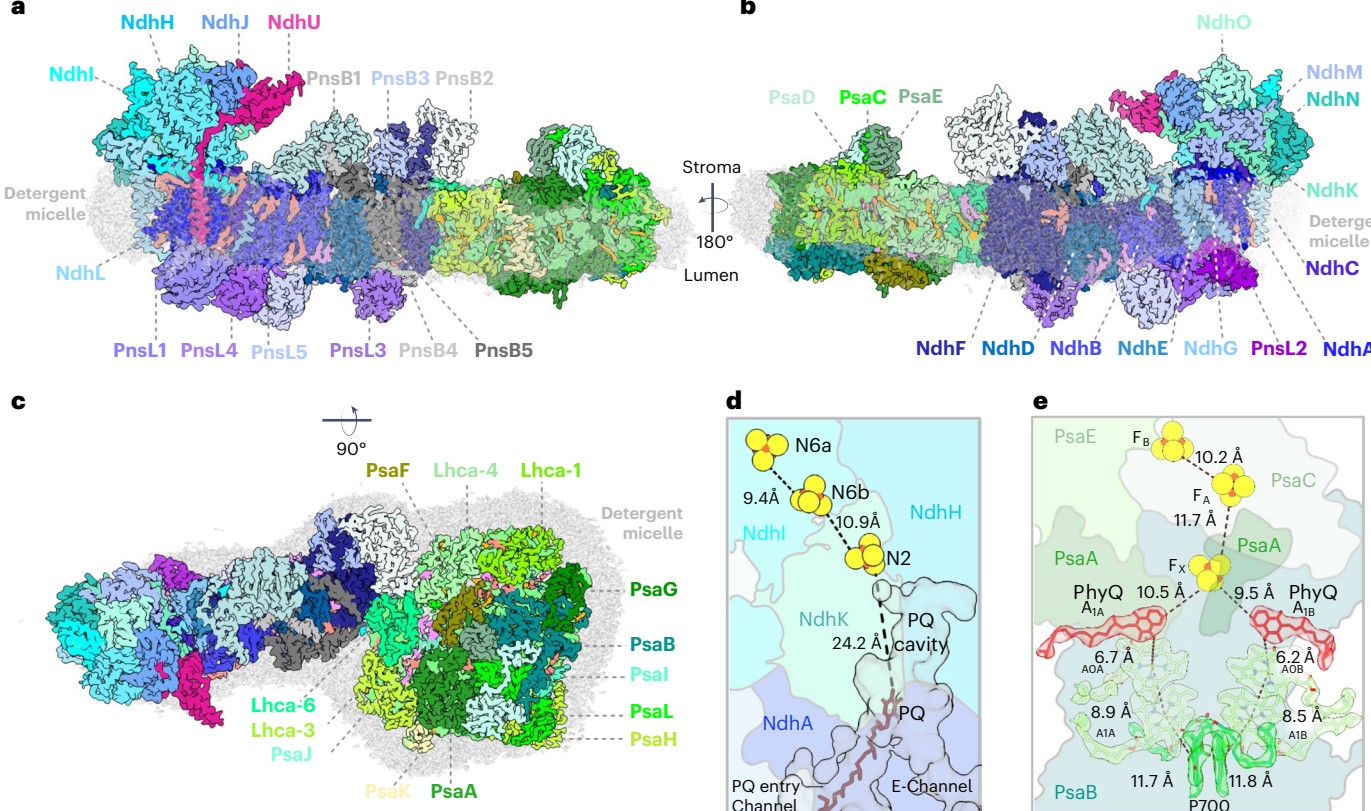

**Fig. 1 | Cryo-EM structure of the NDH–PSI–LHCI-2 supercomplex from** ***S. oleracea.*** **a**–**c**, Lateral (**a**,**b**) and top (**c**) views of the cryo-EM map of the supercomplex (contour level: 0.38). Shades of cyan (module SubA, NdhH–NdhO) and magenta (module SubE, NdhU) indicate the NDH peripheral arm. The membrane arm is drawn in shades of blue (module SubM, NdhA–NdhG). Lumenal SubL subunits (PnsL1–PnsL5) are shown in shades of purple and subunits of the stromal SubB (PnsB1–PnsB5) in shades of gray. PSI–LHC-2 is colored in shades of green. Cryo-EM densities of lipids are salmon (PG), violet (MGDG), pink (SQDG) or green (DGDG). Carotenoids are orange and chlorophylls are bright green. The detergent belt is transparent gray. **d**, The peripheral arm contains a row of

three 4Fe–4S iron–sulfur clusters (N6a, N6b and N2; Fe, orange; S, yellow) closely spaced for direct electron transfer. N2 is next to the PQ cavity that bifurcates into the PQ entry channel and the E-channel. The distance between the putative PQ molecule bound in the entry channel and N2 is too far for direct electron transfer. **e**, Structure of the PSI cofactors with corresponding map densities (contour level: 0.22). Subunits participating in electron transfer are indicated. PSI contains three 4Fe–4S centers ($F_B$, $F_A$ and $F_X$). The protein scaffold holds the special-pair chlorophylls (P700), pheophytins ($A_{1A/B}$ and $A_{0A/B}$) and PhyQ in two near-symmetrical membrane-spanning branches that converge on Fe–S cluster $F_x$.

## Results

### The NDH–PSI–LHCI-2 supercomplex

Our cryo-EM map of the supercomplex has an average resolution of 3.2 Å (according to local refined map resolutions; Extended Data Fig. 4) but well-ordered regions are locally resolved up to 3 Å (Extended Data Fig. 4). The supercomplex consists of one NDH and one PSI–LHCI unit

in angiosperms (Extended Data Fig. 2). Recent cryo-EM structures of the NDH–PSI–LHCI supercomplex from *Hordeum vulgare*[8], *Arabidopsis thaliana*[9] and maize[9] achieved resolutions of 4.5, 3.9 and 4.4 Å, leaving details of critical subunits, cofactors and indeed the entire peripheral arm largely unresolved (Extended Data Fig. 2b,c).

Here, we present the structure of the NDH–PSI–LHCI supercomplex from *Spinacia oleracea* at 3–3.3-Å resolution (Extended Data Figs. 3b and 4 and Table 2). We refer to this complex as NDH–PSI–LHCI-2 (Extended Data Fig. 2). Our cryo-EM map reveals detailed subunit interactions at the level of side chains. The peripheral redox arm of NDH is well resolved and contains a density near the entry of the PQ cavity that fits a bound PQ molecule. The diverse roles of NDH in electron transfer and proton translocation become clear when we compare our map to recent high-resolution structures of mitochondrial complex I. Understanding the structure and function of NDH and its interaction with PSI in detail has important implications for photosynthesis in plants and their adaptation to changing environmental conditions.

(Fig. 1a–c), interacting directly at the terminus of the transmembrane module of NDH and Lhca6 of PSI–LHCI-2 (refs. 7,9). Our cryo-EM density map revealed 26 of the 29 NDH subunits[10] and all 16 subunits of the PSI–LHCI-2 complex[11], including 154 chlorophylls, 37 carotenoids and 46 lipids. NDH is organized into five modules: SubA, SubB, SubE, SubL and SubM. Modules SubA (NdhH–NdhO) and SubE (NdhU) form the peripheral arm. In the SubE module, which is specific to oxygenic photosynthesis, we identified subunit NdhU, while NdhS, NdhT and NdhV[12–15] were absent.

The membrane arm extends from SubB (PnsB1–PnsB5) on the stromal side to SubL (PnsL1–PnsL5) on the lumenal side. These subunits are unique to chloroplast NDH and absent in cyanobacteria[10,16–20]. On the stromal side, PnsB1 interacts with NdhB, NdhD, NdhH, NdhF, PnsB4 and PnsB5 (Fig. 2a–e). PnsB3 has four conserved cysteine residues (Extended Data Fig. 5d) expected to bind an iron–sulfur cluster[21], although we did not find an Fe–S density in this position.

The peripheral arm of plant NDH binds three iron–sulfur clusters precisely spaced for efficient electron transfer. NdhI coordinates two 4Fe–4S clusters and NdhK coordinates one (Fig. 1d). PQ binds in the PQ cavity at the junction between the peripheral and membrane arms (Fig. 1d).

Spinach PSI–LHCI-2 closely resembles the *Pisum sativum* PSI–LHCI complex[22,23], consisting of 12 core subunits (PsaA–PsaL) and four LHCI antenna complexes (Lhca1, Lhca3, Lhca4 and Lhca6; Fig. 1b,c). As observed in NDH–PSI–LHCI-2 supercomplexes from other plants[8,9],

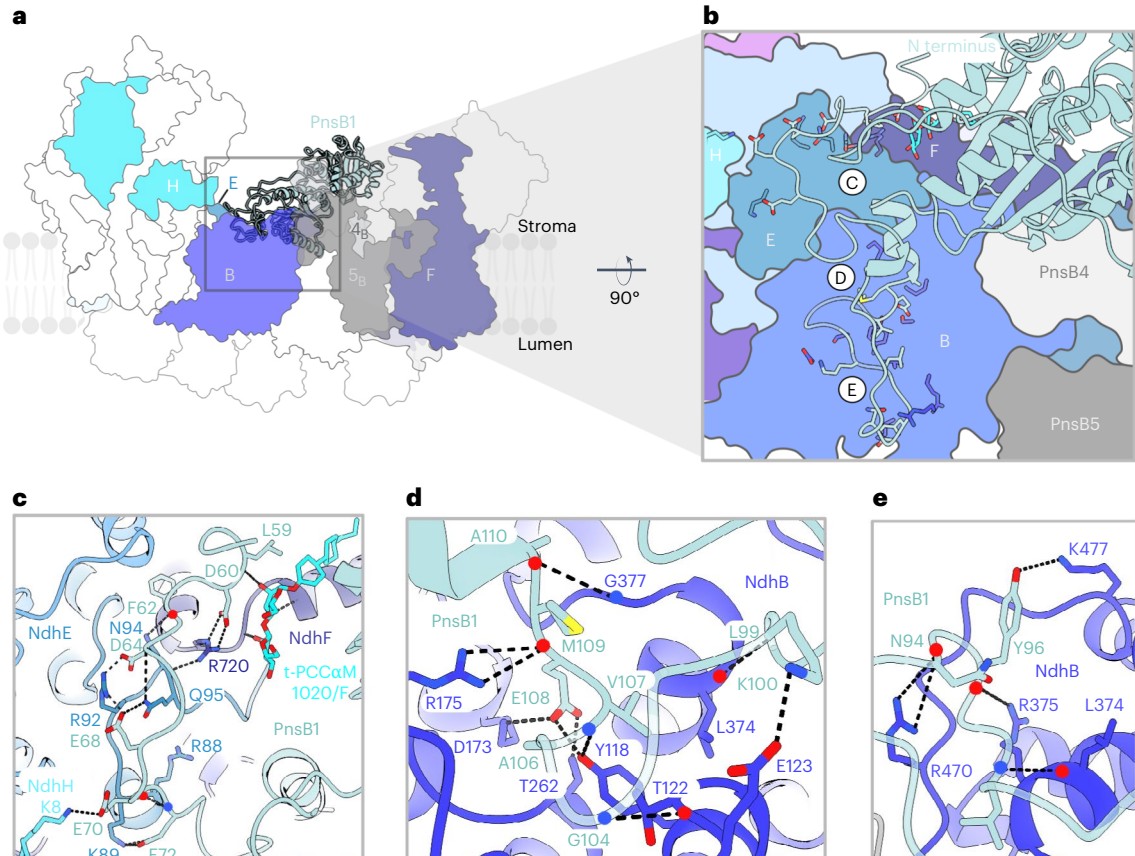

**Fig. 2 | The long N-terminal loop of PnsB1 interacts with NdhB residues likely to be involved in proton translocation. a**, Lateral view of the supercomplex. The subunits interacting with PnsB1 are color-coded as in Fig. 1. PnsB1 is shown as a ribbon; other subunits are shown as an outline. The highlighted region illustrates the points of contact between the long N-terminal loop and the subunits indicated. **b**, Main contact points between the PnsB1 N-terminal loop and the subunits below. Each group of electrostatic interactions is marked with a letter and shown in the corresponding panels (**c–e**).

Lhca6 interacts with NDH, replacing Lhca2 (Fig. 1a–c). The PSI core subunits, PsaA and PsaB, form a conserved heterodimer[22–24], with the reaction center situated at the interface of their C termini. This assembly includes two membrane-spanning branches, A and B, each featuring one chlorophyll P700, two pheophytins ($A_{1A/B}$ and $A_{0A/B}$) and a phylloquinone (PhyQ) (Fig. 1e). The terminal electron acceptors are three 4Fe–4S clusters ($F_X$, $F_A$ and $F_B$), which subsequently reduce Fd. $F_A$ and $F_B$ are bound to PsaC, while $F_X$ is coordinated by four pseudosymmetrical cysteine residues from PsaA and PsaB (Extended Data Fig. 5c). On the stromal side, peripheral subunits PsaC, PsaD and PsaE form the Fd-docking site. PsaG and PsaI are unique to angiosperms and algae[25]. PsaF, PsaA and PsaB together create a potential docking site for the soluble electron transfer protein plastocyanin[26,27].

In our PSI–LHCI-2 density map, we observed 154 chlorophylls, matching the count in PSI–LHCI-2 from *A. thaliana* but exceeding the 148 chlorophylls in PSI–LHCI-2 from *H. vulgare* (Supplementary Fig. 1a,c). Additionally, PSI–LHCI-2 from *S. oleracea* has more chlorophylls than PSI–LHCI-1 from both *A. thaliana* (152) and barley (148) (Supplementary Fig. 1d,e). This discrepancy is most apparent in subunit Lhca5, which in PSI–LHCI-1 interacts with NDH (chlorophylls labeled a, b and z in Supplementary Fig. 1d,e), suggesting that differences in pigment content reflect the variation in subunit composition and that the presence or absence of particular chlorophylls might be required for energy transfer in PSI–LHCI-1.

### NDH–PSI–LHCI-2 interaction

In the spinach supercomplex, Lhca6, NdhF and PnsB5 stabilize the connection between NDH and PSI–LHCI-2 (Fig. 3a,b). Numerous electrostatic, polar and hydrophobic interactions contribute to supercomplex formation and stability, primarily on the stromal side (Fig. 3c–h). Lhca6 interacts with NdhF mainly through its transmembrane helix 2 (TMH2)–TMH3 loop (Fig. 3c–e). One electrostatic interaction was found on the lumenal side between the TMH1–TMH2 loop of Lhca6 and the N-terminal helix of NdhF (Fig. 3f). The extensive N-terminal stromal loop of PnsB5 engages with both PnsB1 (Extended Data Fig. 6a,b) and NdhD (Extended Data Fig. 6a,c,d), passing above NdhF (Extended Data Fig. 6a,c–e) and forming a hook around the N-terminal stromal loop of Lhca6 (Fig. 3g and Extended Data Fig. 6e). A detergent molecule, likely replacing a lipid, appears to reinforce the connection between TMH3 of Lhca6, the N-terminal stromal loop of PnsB5 and the α-helix motif linking TMH1 and TMH2 of NdhF (Fig. 3h). The loop connecting β-strands 4 and 5 of PnsB2 closely approaches the TMH2–TMH3 loop of Lhca6 (Fig. 3c). Contrary to expectations[28], we did not observe direct interactions of NDH PnsB2 and PnsB3 with PSI–LHCI-2.

### The NdhU subunit

The local resolution of the peripheral arm of spinach NDH was 3.3 Å, enabling detailed mapping of subunit interactions at the level of individual side chains in the SubA and SubE modules. Biochemical assays[12,29,30] and cryo-EM structures of cyanobacterial NDH[31,32] both revealed that SubE is necessary for the interaction of the complex with Fd. In addition to the subunits found in cyanobacterial NDH, we discovered a previously unresolved subunit (Fig. 4a and Extended Data Fig. 2) that we identified on the basis of its resolved side-chain sequence (Fig. 4a) as NdhU of the SubE module[12]. AlphaFold[33] models for NDH subunits previously identified by mass spectrometry[12–14] (Fig. 4b,c) suggested that

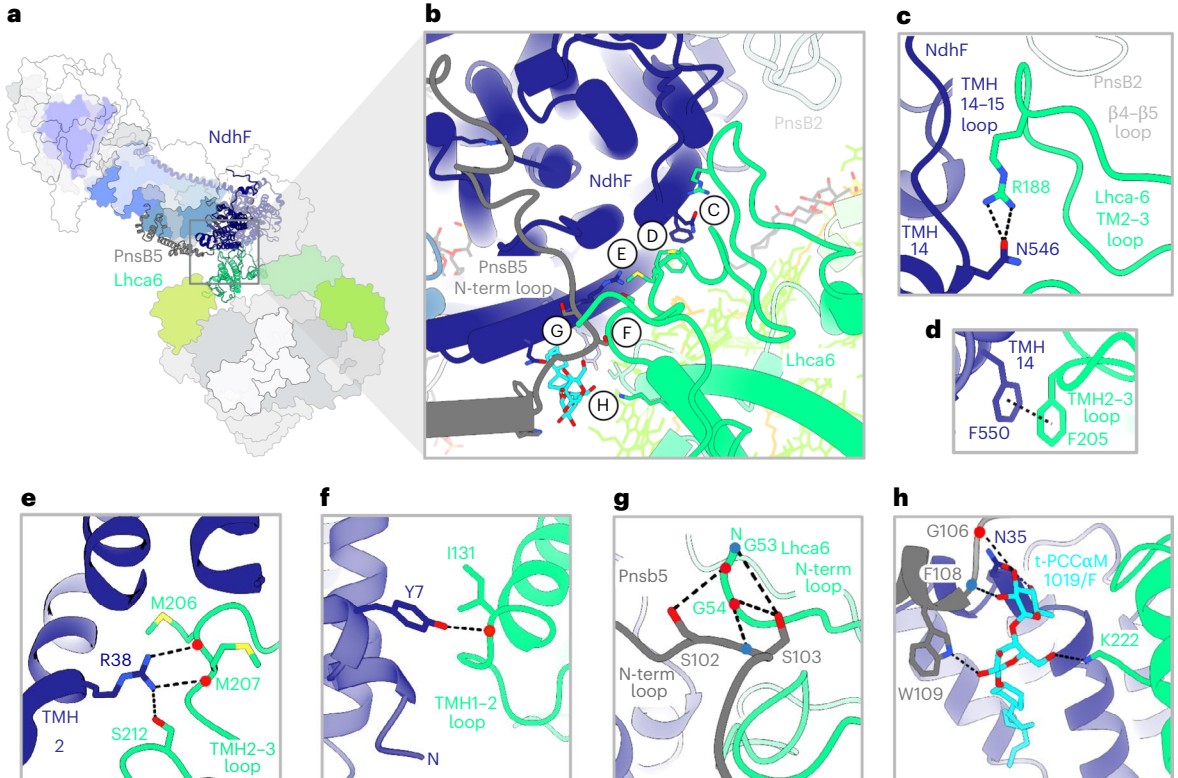

**Fig. 3 | The N-terminal loop of PnsB5 is central to supercomplex formation and stability. a**, Top view of the supercomplex. The subunits connecting the two complexes are shown as ribbons (NdhF, dark blue; PnsB5 dark gray; Lhca6, spring green) while the other subunits are shown as outlines. Key subunits are color-coded as in Fig. 1. **b**, Highlighted region illustrating the points of subunit contacts. **c–h**, Interactions in detail.

either NdhT (Fig. 4c, yellow) or NdhU (Fig. 4c, magenta) was compatible with this new EM density (Fig. 4e). Both models featured a J-shaped domain that was visible in the map but side-chain densities of aromatic residues identified the subunit as NdhU unambiguously (Fig. 4d–h).

NdhU has a flexible N-terminal loop and a J-shaped domain composed of four α helices (α1–α4), followed by a short helix (α5) (Extended Data Fig. 7a). An extended loop connects α5 to the transmembrane domain (Extended Data Fig. 7a). NdhU is centrally placed in the SubE module, establishing electrostatic interactions and salt bridges with subunits NdhI, NdhH, NdhK and NdhJ on the stromal side and potentially with NdhA on the lumenal side (Extended Data Fig. 7e). Its long N-terminal loop and J-shaped domain interact with NdhJ forming an aromatic cluster with NdhK, anchoring it to the peripheral arm (Extended Data Fig. 7b,c). Additionally, the loop connecting the soluble part to the NdhU TMH interacts with the surface of the SubA module (Extended Data Fig. 7d,f).

## Bound lipids

We modeled a total of 46 lipids into distinct densities in the spinach supercomplex map, of which 23 are in the NDH membrane arm and 23 are in PSI–LHCI-2 (Extended Data Fig. 8a,b). In NDH, we found densities for 14 phosphatidyl glycerol (PG), 5 monogalactosyl diacylglycerol (MGDG) and 3 sulfoquinovosyl diacylglycerol (SQDG) lipids (Extended Data Fig. 8a) but, surprisingly, no digalactosyl diacylglycerol (DGDG). Most lipids are associated with transmembrane subunits (Fig. 5, middle) and many fill gaps between neighboring subunits, acting as a hydrophobic adhesive (Fig. 5a–i). The negatively charged PG head groups connect loops and helices (Fig. 5a–c,g–i). Two adjacent PG molecules link the N-terminal loop of NdhL to NdhA (Fig. 5b), as in cyanobacterial NDH[32] (Supplementary table 1a). Nested within the NDH core, we detected three well-defined SQDG molecules at the juncture between NdhB, NdhD and PnsB1 (SQDG 522/B; Fig. 5d), between NdhD

and NdhF (SQDG 806/D; Fig. 5e) and between NdhF and PnsB5 (SQDG 1015/F; Fig. 5f). SQDG 1015/F, together with two PGs and a molecule of β-carotene[8,32,34] (BCR 1003/D), connect NdhD, NdhF, PnsB5 and PnsB4 (Fig. 5f).

Of the 23 lipids modeled into the PSI–LHCI-2 complex, PG is the most abundant, followed by MGDG (Extended Data Fig. 8b). The lipid distribution within PSI–LHCI-2 is striking. While PG and DGDG are located at the interface between LHCIs and PSI or within the PSI core (Extended Data Fig. 9, middle), MDGD and SQDG are confined to the border between the LHCs and PSI (Extended Data Fig. 9, middle; violet and pink densities respectively). At the interface with Lhca1, Lhca6 and Lhca3 (Extended Data Fig. 9a,c,f,h), a prominent string of lipids connects the LHCI belt to PSI. MGDG in this region is consistent with the crystal structure of *P. sativum* PSI–LHCI[22,23] (Supplementary table 1b). Some PG lipids bind closely to chlorophyll, creating a ligand for the central $Mg^{2+}$ (Extended Data Fig. 9b–e). The symmetrical arrangement of one PG (PG 1064/a) and one DGDG (DGDG 850/b) in the PSI core is conserved across species (Extended Data Figs. 8b and 9e).

## The PQ-binding pocket

At the juncture of the membrane arm and the peripheral arm, the HOLLOW software tool[35] revealed a bifurcated cavity. This cavity comprises the PQ entry channel, E-channel and PQ-binding pocket, closely resembling the homologous Q entry channel, E-channel and Q-binding pocket of complex I (Figs. 1d and 6a,b). Within the PQ entry channel, we observed a distinct nonprotein density of approximately 26 Å in length (Fig. 6b, red) in a position that corresponds to the shallow site in mitochondrial complex I[36,37]. Guided by previously published structures of cyanobacterial NDH[32] and complex I[36,38], we modeled a PQ molecule in the density (Fig. 6c). The PQ sits at the entrance of the cavity surrounded by several charged residues, interacting with Y259

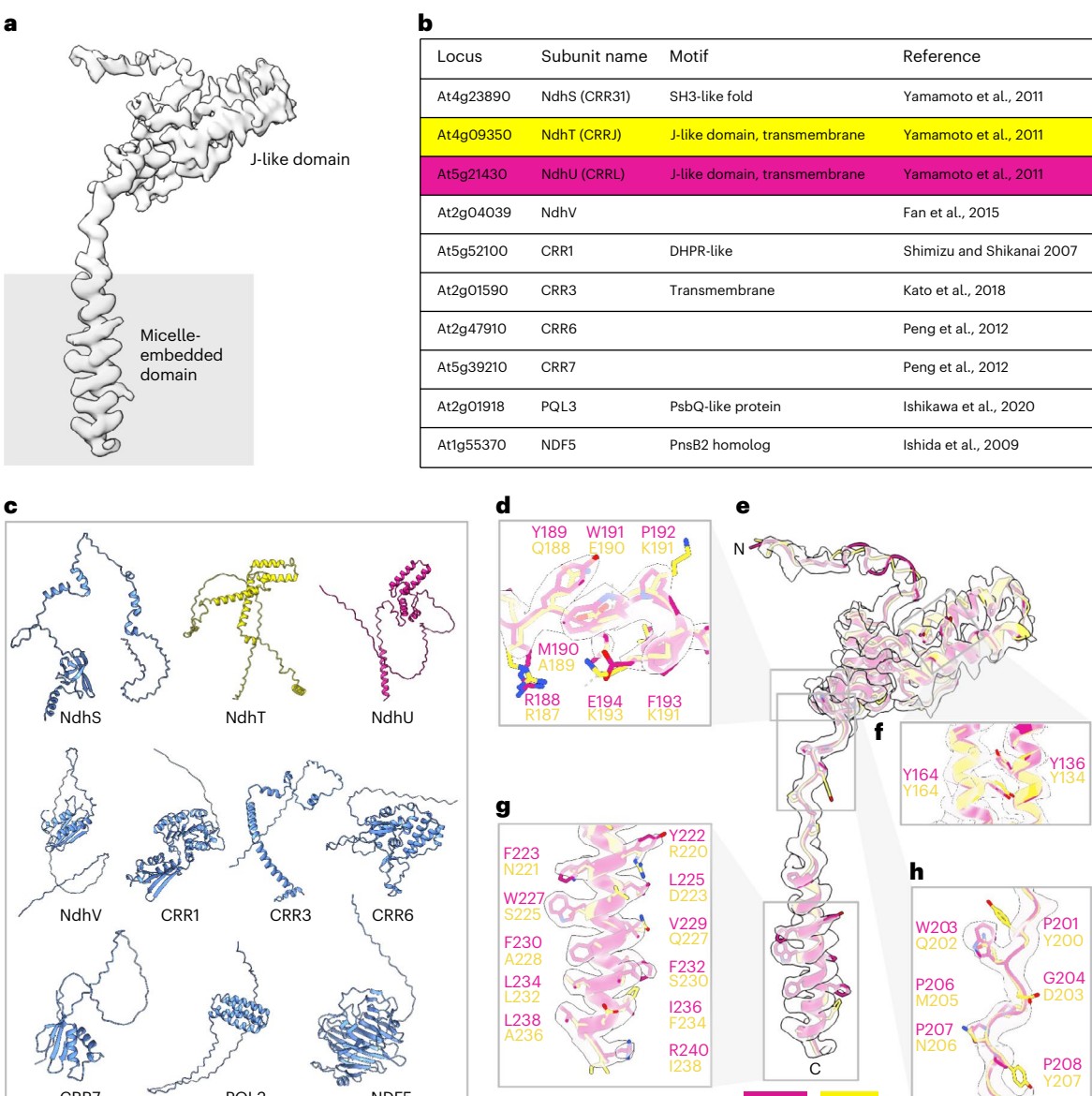

**Fig. 4 | Identification of NdhU from its cryo-EM density. a**, Cryo-EM map of the unidentified subunit. **b**, List of nuclear-encoded subunits and protein factors with established interactions with NDH observed in the model organism *A. thaliana*. **c**, Panel with the computational models of the proteins listed in **b**. The designated subunits were produced with AlphaFold[33] from *S. oleracea* input sequences. **d–h**, On the basis of the AlphaFold predictions and the results from Yamamoto et al.[12], we attempted to fit both NdhT and NdhU into the cryo-EM map shown in **a**. Side chains of NdhU (magenta) fit the density map well, firmly establishing the identity of this subunit. The cryo-EM map shown in **a,e,g** is drawn at a contour level of 0.3. In **d,f,h** the contour level is 0.45 to show the side chains more clearly.

of NdhA through its carbonyl group (Fig. 6c). The PQ head group is directed toward the N2 Fe–S cluster at an approximate distance of ~24 Å (Fig. 1d), too far for direct electron transfer. Although we cannot rule out t-PCCαM (4-*trans*-(4-*trans*-propylcyclohexyl)-cyclohexyl α-maltoside), the detergent used for solubilizing and purifying the supercomplex, the density in the PQ pocket looks too long and bulky for a detergent molecule and PQ fits it almost perfectly (Fig. 6c).

The PQ-binding pocket is formed by subunits NdhA, NdhH and NdhK (Fig. 6a,b). As in cyanobacterial NDH[32], the PQ pocket is lined by the four-helix bundle and the loop connecting the first two β-strands (β1–β2 loop) of NdhH. The highly conserved Y72 residue approaches Fe–S cluster N2 closely (Fig. 6b). The PQ pocket is further lined by the second helix (α2) and the loops α1–α2 and α3–α4 of NdhK and the long loop connecting TMH5 and TMH6 of NdhA. This loop engages in numerous intrasubunit and intersubunit interactions, including three

salt bridges between NdhA and NdhH (Fig. 6d). Note that the TMH5–TMH6 loop is not fully resolved in PQ-bound cyanobacterial NDH[32].

The special confinements of the PQ pocket are lined with bulky hydrophobic residues (Fig. 6b). In the NdhH β1–β2 loop, the highly conserved residues H23 and H19 are oriented outward (Fig. 6b), leaving the path to Y72 open. The tyrosine residue (Y39) in the NdhK α1–α2 loop is not conserved in complex I but present in both plant and cyanobacterial NDH.

The PQ-binding pocket is formed by subunits NdhC, NdhI, NdhJ, NdhL, NdhM, NdhO and NdhU (Fig. 6a,e), of which NdhC, NdhI, NdHL and NdhU link the membrane arm to the hydrophilic arm of NDH. The long stromal loop connecting the first and second TMHs of NdhC (TMH1–TMH2 loop; Fig. 6e) connects to NdhA and the soluble subunits NdhK, NdhH and NdhM with multiple salt bridges, reinforcing the interactions between the two arms of NDH (Fig. 6e).

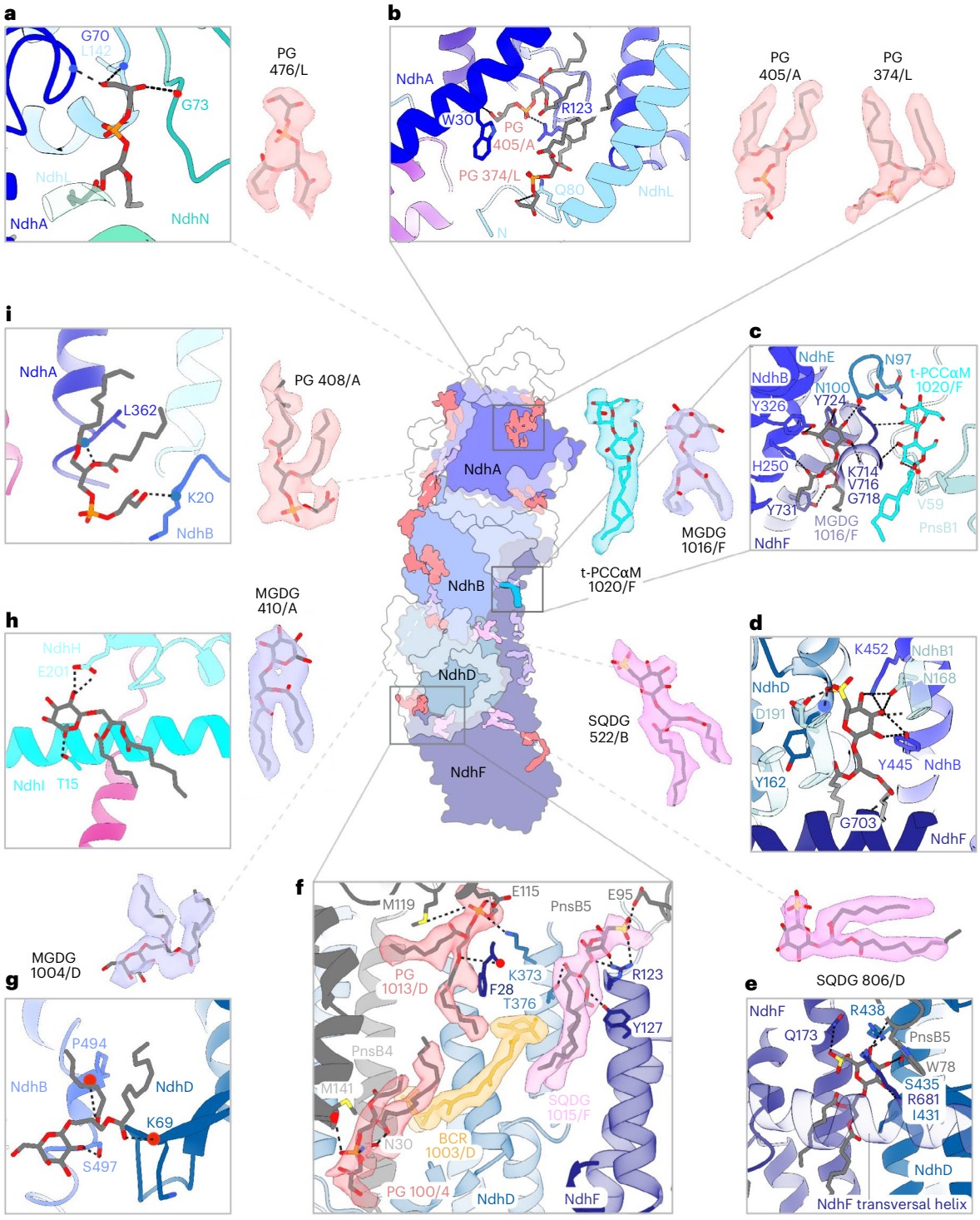

**Fig. 5 | Lipids in the NDH membrane arm.** Middle, subunits of the NDH membrane arm in outline, with NdhA, NdhB, NdhD and NdhF color-coded. Stromal and lumenal subunits were removed for clarity. Lipid densities are salmon (PG), violet (MGDG) or pink (SQDG). Detergent (t-PCCαM (A1H1M)) density is turquoise. **a**–**i**, Several lipids interact with more than one subunit, presumably enhancing supercomplex stability.

## The transmembrane proton pathway

The transmembrane module of NDH comprises seven subunits (NdhA–NdhG). Of these, NdhA, NdhC, NdhE and NdhG form the E-channel[37] (Fig. 7a,b) and NdhB, NdhD and NdhF correspond to the antiporter-like subunits of complex I[38] (Fig. 7a). Together, they constitute the four proton-pumping units known from complex I[5,37,39]. Each antiporter-like subunit contains two discontinuous helices (TMH7 and TMH12, Fig. 7a) that are integral to the formation of conserved internal cavities within the membrane arm (Fig. 7a, gray surfaces).

As in cyanobacterial NDH[32], we observed that NdhA, NdhH and NdhK subunits enclose a branched cavity (Fig. 7a, gray surface). The primary branch lined by NdhA helices forms the access channel for PQ toward the PQ entry channel and the PQ-binding site in subunits NdhH and NdhK (Fig. 7a). The second branch forms the E-channel[37], a passage extending approximately 32 Å through NdhA toward subunits NdhC, NdhG and NdhE in the membrane arm (Fig. 7a,b). The E-channel is defined by multiple glutamate residues (Fig. 7b), most of which are strictly conserved between NDH and complex I (Supplementary

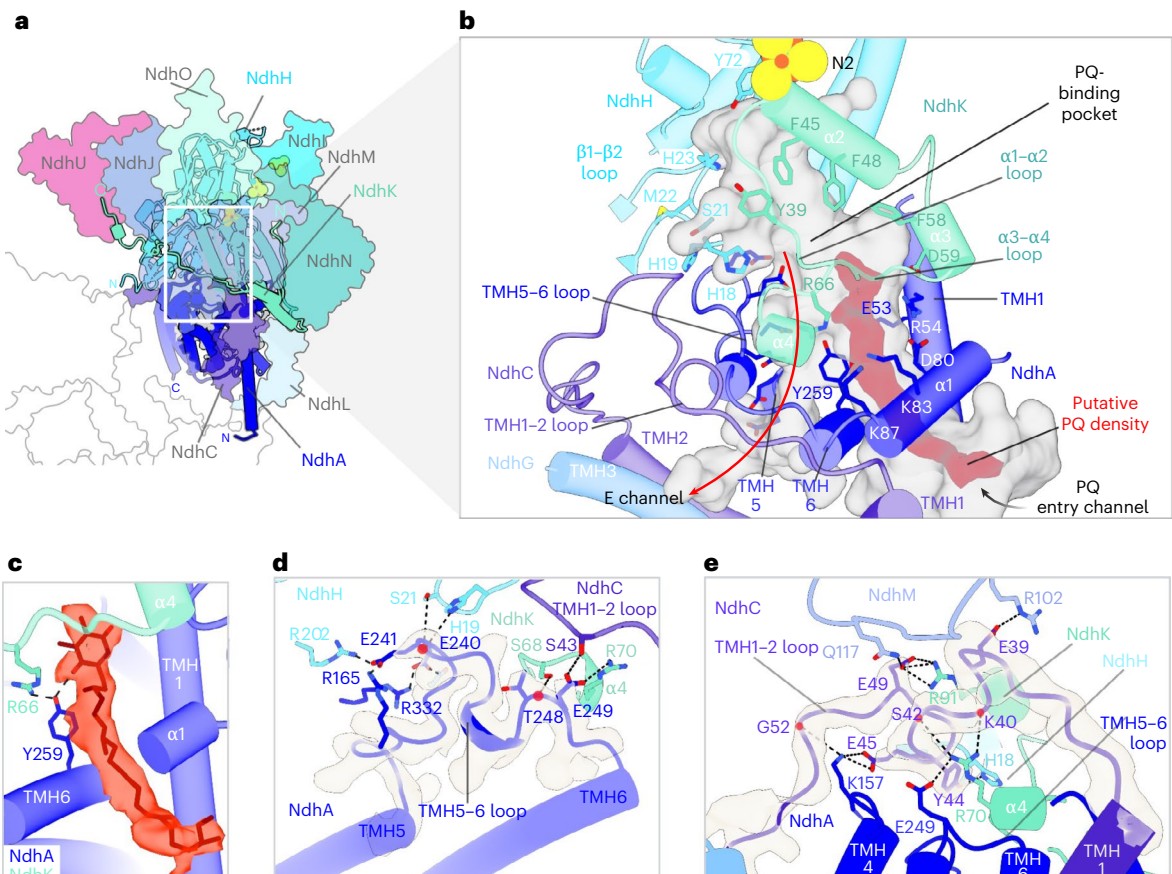

**Fig. 6 | The PQ-binding pocket. a**, Lateral view of the peripheral arm with subunits forming the PQ chamber (NdhH, NdhK and NdhA) shown as cylinders and surrounding subunits in outline. **b**, Secondary-structure elements defining the PQ pocket. The PQ density is red and the cavity is shown as a gray surface. **c**, Close-up view of the PQ molecule modeled into the density (red) visualized at a contour level of 0.23. PQ interacts with Y259. **d**, TMH5–TMH6 loop of NdhA and the density map (light gray) at a contour level of 0.4. Residues making intrasubunit and intersubunit interactions are shown as sticks. **e**, Salt bridges formed between the NdhC TMH1–TMH2 stromal loop and surrounding subunits. The density of the NdhC TMH1–TMH2 loop is shown in gray at a contour level of 0.4. Backbone carbonyl groups involved in side-chain interactions are highlighted red in **d**,**e**.

Figs. 2–5). Among these residues, the functionally important glutamates of complex I[40] $E148_{ND1\_At}$ ($E174_{NdhA}$) and $E197_{ND1\_At}$ ($E227_{NhdA}$) are conserved, while $E232_{ND1\_At}$ is substituted by a serine in NDH ($S262_{NdhA}$; Fig. 7b and Supplementary Fig. 2h, yellow arrows). In the E-channel, protons injected from NdhA initially encounter the NdhG subunit (homologous to ND6 of mitochondrial complex I; Supplementary Fig. 3). In complex I, TMH3 of ND6 forms a $\pi$-bulge that opens and closes the aqueous passage through the membrane arm[37,41]. Our map shows no evidence of a $\pi$-bulge in the corresponding helix (Fig. 7c). At this position the E-channel is continuous but the potential proton passage is interrupted at the antiporter-like subunit NdhB (Fig. 7b). As in complex I[37,42–44], the subunits responsible for proton translocation define a chain of buried, charged residues along the potential proton passage (Fig. 7a, gray surfaces) from the E-channel to the half-channels in NdhB, NdhD and NdhF (Supplementary Fig. 6–8).

We mapped three shallow half-channels at the stromal side above the discontinuous helix TMH7 and the broken helix TMH8 of NdhB, NdhD and NdhF (Fig. 7a; labeled 1, 2 and 3 respectively). A network of conserved charged and polar residues connects the stromal half-channels to the buried half-channels[37,42–44]. The chloroplast subunit PnsB1 sits exactly above the half-channel on the NdhB stromal site (Fig. 7d). The PnsB1 long N-terminal loop rich in charged residues defines part of a funnel leading to the conserved residues $E258_{NdhB}$, $K321_{NdhB}$ and $R332_{NdhB}$ that are implicated in proton translocation in complex I[37,39,42,44]. A t-PCCαM detergent molecule (A1H1M 1020/F; Fig. 7d) and one MGDG (MGDG 1016/F) contribute to the formation of

the funnel by blocking the lateral exit, interacting with NdhB, NdhE, the transversal helix of NdhF and PnsB1 (Fig. 5c).

We observe a potential proton channel at the interface between NdhB and NdhD, which we refer to as the NdhB–NdhD channel (Fig. 7a,e–g). The NdhB–NdhD channel is lined by TMH12–TMH13 of NdhB and TMH5–TMH6 of NdhD. It is blocked by two negatively charged lipids (Figs. 7e–g and 5e) in positions that are conserved from complex I to NDH (Supplementary Table 1a). This wide tunnel opens toward the stroma through a network of charged residues (Fig. 7e,g) and is closed on the lumenal side by two tryptophan side chains (Fig. 7e). Surface analysis indicates that the subunits above the NdhB–NdhD channel together form a charged funnel, potentially facilitating proton translocation (Fig. 7g). A triad of nonpolar residues shields the passage from the charged residues in the proton translocation pathway (Fig. 7f). It is worth noting that all our maps of NDH indicate nonprotein densities within the membrane arm channels (Extended Data Fig. 10, cyan), which are most likely water molecules.

## Discussion

Our analysis of the spinach NDH–PSI–LHCI-2 supercomplex reveals surprisingly close similarities but also some unique features compared to complex I. The map enables a precise identification of NDH modules at the side-chain level, especially in the previously unresolved peripheral arm. SubA extends toward the stroma. While some of its subunits (NdhH–NdhK) share homology with those that form the complex I Q module (Extended Data Fig. 1), others (NdhL–NdhO)[16] are confined to

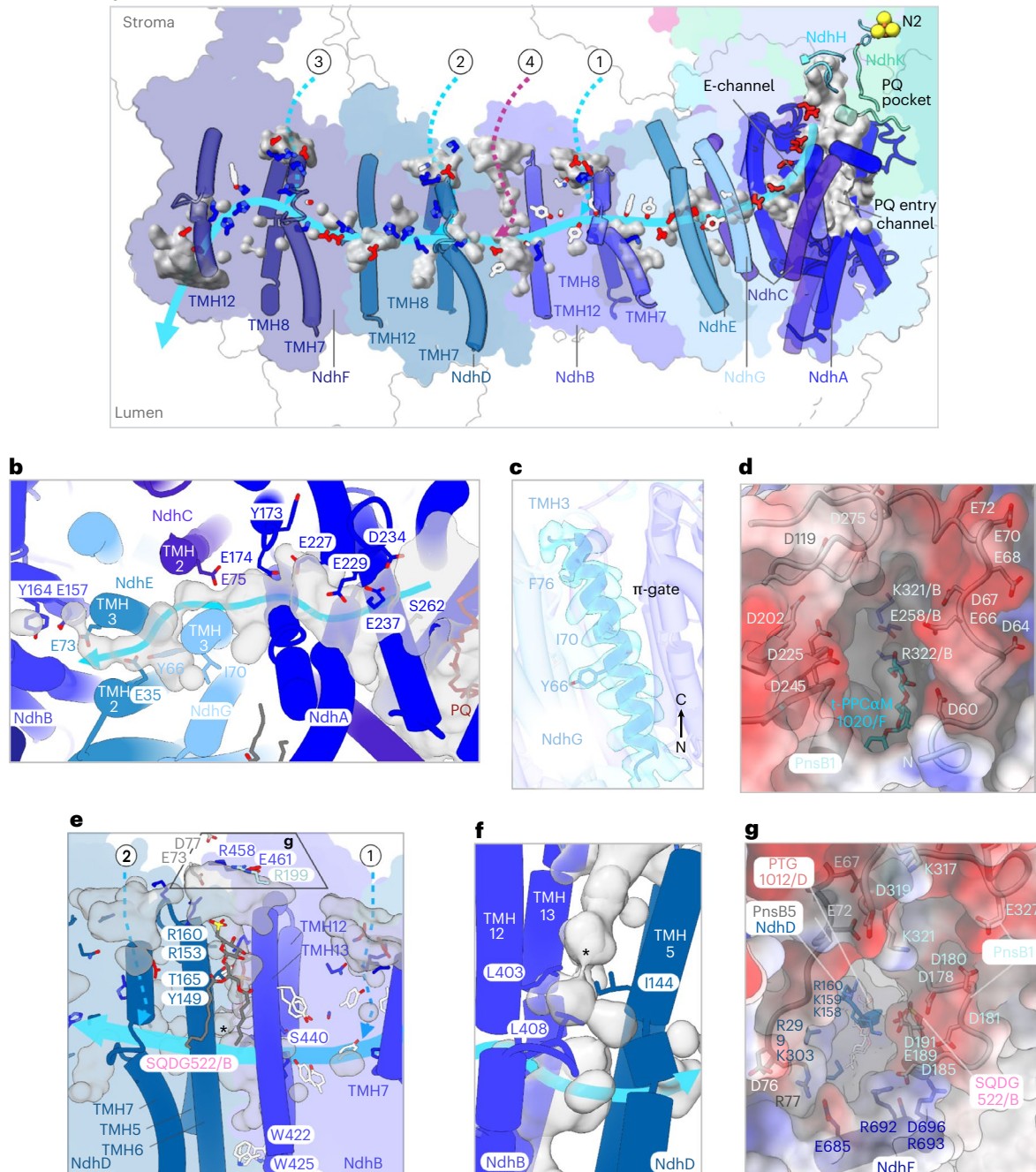

**Fig. 7 | Proposed proton transfer path in the membrane arm of _S. oleracea_ NDH. a**, Side view of NDH. From right to left, NdhA helices and NdhH and NdhK loops contributing to the formation of the PQ-binding pocket (gray surface), NdhC TMH1–TMH2, NdhG TMH3, NdhE TMH2–TMH3 and discontinuous TMHs of NdhB, NdhD and NdhF are shown as solid cylinders. Other subunits are shown as outlines. The 4Fe–4S N2 cluster is indicated. Charged residues of NdhA–NdhG and residues involved in proton translocation that are conserved between spinach NDH and _A. thaliana_ complex I are shown as stick models and are color-coded. Positively charged residues are blue, negatively charged residues are red and key aromatic residues are white. PQ pocket, voids, channels and depressions within NDH that are likely to be relevant for proton translocation are shown as gray surfaces. The light-blue arrow delineates a continuous hydrophilic path from the Q-binding site to NdhF. Stromal half-channels above NdhB, NdhD and NdhF are indicated as dotted cyan arrows and labeled 1–3; the channel between NdhB and NdhD (magenta) is labeled 4. **b**, Top view of the E-channel with the

relative map of voids presumably occupied by water molecules. Key residues are labeled. **c**, TMH3 of NdhG. Conserved residues in _A. thaliana_ complex I important for proton translocation are shown in stick representation. The cryo-EM map (transparent cyan) is at a contour level of 0.4. **d**, Top view of the channel entry above NdhB shown as a surface (1 in **a**). Subunits of to the entry channel are shown as electrostatic potential surfaces (red, negative; blue, positive). The PnsB1 N-terminal loop is shown in cartoon representation. **e,g**, Side (**e**) and top (**g**) views of channel NdhB-D (4 in **a**). Polar residues defining the putative hydrophilic channel are shown as sticks. In **e**, the asterisk indicates the constriction point shown in **f**. **f**, Close-up view of the constriction in channel NdhB–NdhD. As in **e**, the asterisk indicates the constriction point. In **g**, elements from PnsB1, PnsB4 (not visible), PnsB5, NdhB and NdhD form a funnel, depicted as electrostatic potential surfaces (negative, red; positive, blue) above the stromal entry. The subunits involved in forming the funnel are shown in cartoon representation. Charged residues are represented as sticks.

organisms involved in oxygenic photosynthesis. Chloroplast-specific subunits SubB and SubL are thought to stabilize NDH, enhancing its activity[27,45]. At this stage, their precise function in proton translocation and photosynthesis remains unclear, necessitating further research. Previously, the extended N-terminal loop of PnsB1 was assigned to an unknown protein[8] (Supplementary Fig. 9a,f). We now find it interacts with stromal-exposed charged residues of NdhB, suggesting a role in proton translocation across the thylakoid membrane (Figs. 2c–e and 7d). Even though we isolated the supercomplex under reducing conditions, some cysteines in subunit PnsB3 formed disulfide bonds (Extended Data Fig. 5a,b,d,e), in contrast to the *Arabidopsis*[9] and barley[8] supercomplexes that were purified without reducing agents but show an Fe–S density in this position.

The 4Fe–4S clusters in the peripheral arm of plant NDH resemble those in cyanobacterial NDH[31,32,34,46]. Notably, their arrangement mirrors that of the clusters N6a, N6b and N2 in the peripheral arm of complex I[47–49], suggesting a pathway for electron transfer from Fd to PQ (Fig. 1d) and underscoring their functional importance in photosynthetic electron transfer.

Our PSI–LHCI-2 contains the same number of chlorophylls (154) as PSI–LHCI-2 from *A. thaliana*[9] but more than PSI–LHCI-2 from *H. vulgare*[8] (148) (Supplementary Fig. 1a,c). We attribute this difference first and foremost to the substantially higher resolution of our map (EMD-19248, 3.09 Å) compared to barley[8] PSI–LHCI-2 (EMD-31350, 3.88 Å). At a resolution of around 3.9 Å, pigment densities are not very clear and can be easily missed or mistaken. Second, both PSI–LHCI-1 and PSI–LHCI-2 complexes from barley[8] lack the subunit PsaG (Supplementary Fig. 1c), which, as in our spinach structure and in *A. thaliana*[9], contains several chlorophylls and carotenoids. Certain chlorophylls are observed only in barley[8] (x and y in Supplementary Fig. 1a,d). These disparities in the number and position of chlorophylls in barley may be species specific. This may reflect the evolutionary distance between barley, a monocotyledon, and spinach or *A. thaliana*, which are dicotyledons.

Lhca6 of PSI is known to be required for interaction with NDH[6,50]. Our study offers clear evidence that Lhca6 interacts with the SubB module of NDH, whereas earlier conclusions[28,51] were speculative. Some studies[7–9] found that NDH can bind a second PSI–LHCI assembly (that is, PSI–LHCI-1), through interactions of the PnsB1 β3–α5 and β5–β6 loops and the Lhca5 TMH2–TMH3 loop (Extended Data Fig. 2b,c and Supplementary Fig. 9b–e). Sequence alignments of spinach PnsB1 and Lhca5 with the corresponding subunits in *A. thaliana* and *H. vulgare* showed no notable amino acid differences in the interacting interface, ruling out sequence variations as the cause for the observed 1:1 ratio of NDH to PSI–LHCI (Supplementary Fig. 9g–i). The abundance of interactions between Lhca6 and NDH (Fig. 3c–h) potentially stabilizes the formation of the smaller supercomplex, while transient interactions may favor formation of the larger supercomplex with two PSI assemblies[8,9]. Our findings support previous studies indicating that particles with a 1:1 NDH-to-PSI–LHCI ratio predominantly preserve PSI–LHCI-2 (refs. 7–9). PSI–LHCI-2 refers to the position of PSI relative to NDH as illustrated in Extended Data Fig. 2. The higher stability of the smaller supercomplex may be the result of evolutionary adaptation, suggesting that angiosperms evolved to stabilize the NDH complex by forming the Lhca6-dependent supercomplex instead of producing new NDH complexes in mature leaves[28].

Immunoblotting and mass spectrometry consistently detected Lhca6 in the NDH–PSI–LHCI-2 supercomplex[6] but not in PSI monomers from various plants[11], thus suggesting a key role of Lhca6 in supercomplex formation. However, variations in the number of bound PSI–LHCIs may be influenced by environmental factors or leaf maturity[52,53]. Further investigation is needed to determine whether the reversible formation of the supercomplex is dependent on environmental stress, light intensity or leaf maturity.

Of the 23 lipids modeled into the PSI–LHCI-2 complex, PG and MGDG have key roles in photosynthetic activity[54,55] and LHCI stability[56].

Depletion of MGDG in *A. thaliana* seedlings has been shown to disrupt the assembly of the PSI–LHCI complex and the formation of LHCI aggregates[57]. Collectively, these findings underscore the pivotal role of MGDG in PSI–LHCI complex assembly. The direct interaction of PG with chlorophylls may explain the reduced chlorophyll accumulation in the PG-lacking *A. thaliana* pgp1–pgp2 mutant[54]. Moreover, the symmetrical positioning of PG (1064/a) and DGDG (850/b) within the PSI core (Extended Data Fig. 9e) may be important for the assembly of the PSI reaction center, given that PG depletion leads to decreased photosynthetic activity in *A. thaliana*[54,55].

In summary, our detailed insights into NDH–PSI–LHCI-2 supercomplex architecture, subunit interactions and shared characteristics with complex I pave the way for understanding the mechanisms underlying electron transfer and supercomplex formation in photosynthetic organisms. Previously published in vitro assays demonstrated the proton-pumping activity for NDH experimentally[4]. Furthermore, the presence of charged residues in the core subunits of complex I, known to be vital for proton translocation activity[39,58,59], are conserved in NdhA–NdhG of NDH (Supplementary Figs. 2–5). Collectively, our structural findings indicate that chloroplast NDH and mitochondrial complex I function in essentially the same way. This finding holds promise for future advancements in optimizing the photosynthetic process for applications from agriculture to renewable energy.

## Online content

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

## Methods

### Isolation of NDH–PSI–LHCI-2 from spinach leaves

Preparation of thylakoid membranes from young leaves of market spinach (*S. oleracea*) and membrane protein solubilization were carried out as described previously[60]. The suspension was diluted 1:1 with suspension buffer (300 mM sorbitol, 150 mM NaCl, 10 mM Tris-HCl pH 8.0, 0.5 mM MgCl₂, 2 mM KCl, 2 mM DTT and 2 mM PMSF) and sonicated for 120 s on ice with an ultrasonic cell disruptor (Sonifier s-250d digital, Branson Ultrasonics). Grana stacks were sedimented by centrifuging thylakoid suspension for 15 min at ~12,500$g$ (9,000 rpm; JA14.50, Beckman Coulter). Supernatant containing the stromal lamella fraction was carefully removed and diluted 1:1 with extraction buffer (30 mM HEPES pH 7.8, 2 mM MgCl₂, 2 mM KCl, 0.5 mM EDTA, 50 mM NaCl, 1% (w/v) t-PCCαM (Glycon Luckenwalde), 10% glycerol and 2 mM DTT) and incubated for 30 min at 4 °C in dark. Next, the sample was centrifuged for 30 min at 40,000 rpm (Ti45, Beckman Coulter). Supernatant was loaded on a POROS GoPure HQ 50 anion-exchange column (Life Technologies) equilibrated with buffer A (30 mM HEPES pH 7.8, 2 mM MgCl₂, 2 mM KCl, 0.5 mM Na₂-EDTA, 100 mM NaCl, 2 mM DTT and 0.04% (w/v) t-PCCαM) and gradually eluted with 10–60% buffer B (buffer A with 1 M NaCl) for three column volumes using an Äkta Explorer chromatography system (GE Healthcare) at 4 °C (Supplementary Fig. 10a). Peak fractions were pooled and loaded on a clear native PAGE to confirm protein content (Supplementary Fig. 10b). Fractions containing NDH1–PS1 were concentrated to 0.5 ml and loaded on a 16/300 Superose-6 (GE Healthcare Life Sciences) gel-filtration column equilibrated with buffer A (Supplementary Fig. 11a). Samples were eluted with at 0.5 ml min⁻¹. Each fraction was screened with negative-stain EM in an FEI Tecnai Spirit Bio-Twin transmission electron microscope to identify fractions with highest NDH1–PS1 content (Supplementary Fig. 11b). Fractions eluting at ~9–10 ml were concentrated to 1–2 mg ml⁻¹ and used for cryo-EM sample preparation (Supplementary Fig. 11c). Protein concentration was determined using the BCA assay (Thermo Fisher Scientific, Pierce).

### Cryo-EM preparation and EM

First, 3 μl of sample was applied to freshly glow-discharged Quantifoil R1.2/1.3 grids and plunge-frozen in liquid ethane using a Vitrobot (Thermo Fisher Scientific, FEI). Micrographs were recorded in a Titan Krios G2 microscope operated at 300 kV (Thermo Fisher Scientific, FEI) on a K3 direct electron detector in electron counting mode at a pixel size of 0.837 Å, calibrated with apoferritin and an X-ray model as the reference. A total of 55,704 dose-fractionated videos with 40 fractions and a total dose of ~42 e⁻ per A² in a defocus range of −0.7 to −1.7 μm were recorded using EPU and aberration-free image shift (Thermo Fisher Scientific, FEI). Details are shown in Table 1.

### Image processing for single-particle cryo-EM

Images were processed using cryoSPARC[60]. A detailed description of data processing is shown in Supplementary Figs. 12 and 13. Briefly, videos were motion-corrected and contrast transfer function (CTF) parameters were initially estimated as implemented in cryoSPARC. Particle-picking models were manually built using the Topaz[61] wrapper implemented in cryoSPARC and subsequently applied to the datasets. After extraction, particles were subjected to several rounds of two-dimensional classification followed by homogeneous refinement and three-dimensional (3D) classification. The particles from the best classes were merged and, after an initial 3D nonuniform refinement, 3D-refined particles together with a real-space mask around the whole supercomplex were used as inputs for 3D variability analysis[62]. This yielded an initial set of 107,752 particles. Particle coordinates were exported to RELION[63] and underwent 3D refinement, multiple rounds of CTF refinement and Bayesian polishing[63]. Subsequent 3D refinement was carried out in RELION and cryoSPARC, with cryoSPARC producing slightly better results. Next, these particles were subjected to heterogeneous refinement, isolating a subset of 88,276 particles.

The 3D-refined particles with a real-space mask around the peripheral arm were used as inputs for 3D variability analysis to separate particles that showed a well-defined peripheral arm from those where it was damaged (Supplementary Fig. 13c–e). A final set of 38,385 particles was again subjected to nonuniform refinement in cryoSPARC, resulting in a map of 3.34-Å resolution (Extended Data Fig. 3a–c). Local refinement of the various supercomplex components used different masks, one around the NDH peripheral arm (Extended Data Fig. 4a), one around the NDH membrane arm (Extended Data Fig. 4b), one around the border region between NDH and PSI–LHCI (Extended Data Fig. 4g) and one around the PSI–LHCI complex (Extended Data Fig. 4h). Local refinement with these masks improved the overall resolution of the supercomplex to an average of 3.19 Å.

Real-space masks were automatically generated by cryoSPARC. For local refinement, we created a mask base starting from molecular models using the molmap command in UCSF ChimeraX[64]. The mask base was then converted into a mask using volume tools in cryoSPARC. In this process, we set the threshold values chosen in UCSF ChimeraX (which varied from mask to mask), the dilation radius to 0 pixels and the soft padding width to 8 pixels. All other parameters were left at their default settings.

The four locally refined maps were used to generate a final composite map in UCSF ChimeraX[64]. Local resolution estimation was performed in cryoSPARC. All resolutions were estimated according to the Fourier shell correlation (FSC) 0.143 cutoff criterion of two independently refined half-maps[65].

### Model building and analysis

For the initial construction of the model of *S. oleracea* NDH–PSI supercomplex we relied on the structural blueprints provided by the pea PSI–LHCI complex crystal structure (Protein Data Bank (PDB) 4Y28)[22] and the cyanobacteria NDH1L cryo-EM structure (PDB 6KHJ)[32]. Predictions for the plant-specific chloroplastic subunits SubB and SubL of spinach, as well as NdhU, were generated with AlphaFold[33] (Supplementary Figs. 14–16, left). Protein models predicted by AlphaFold are color-coded according to a per-residue model confidence score called the predicted local distance difference test (pLDDT) (Supplementary Figs. 14a, 15a and 16a). The pLDDT score reflects the predicted accuracy of the model in the LDDT at Cα atoms, which measures the reliability of the structural prediction at a local level. The pLDDT score ranges from 0 to 100. Regions with a pLDDT score below 50 may be unstructured. AlphaFold 'predict align error' plots indicate confidence in the relative positioning of two residues, showing domain reliability. Green shades represent the expected distance error (0–31 Å), with the color at ($x, y$) reflecting the predicted error in residue $x$'s position when aligned to residue $y$ (Supplementary Figs. 14–16, right).

Subsequently, the models were initially docked into the various locally refined maps using UCSF ChimeraX[64]. Where necessary, amino acid residues of specific subunits were modified on the basis of the spinach sequence in Coot[66]. Initial model fitting was conducted using ISOLDE[67] and iterative refinement was performed with Coot in conjunction with PHENIX[68]. Final model fitting was performed using the composite map (Extended Data Fig. 3d; EMD-51527) in Coot in conjunction with PHENIX. The quality of the models was assessed with MolProbity[69] (Table 1).

To identify tunnels, cavities and voids in the NDH complex, we used the software tool HOLLOW[35] with default values (that is, probe radius set to 1.4 Å and grid spacing set to 0.5 Å). Sequence alignments shown in Supplementary Fig. 9 were performed with ClustalOmega[70]. Figures showing molecular structures and density maps were created with UCSF ChimeraX[64]. Graphical elements and icons were created with BioRender.com.

### Reporting summary

Further information on research design is available in the Nature Portfolio Reporting Summary linked to this article.

**Table 1 | Cryo-EM data collection, refinement and validation statistics**

| | Dataset 1 | | | Dataset 2 | | |
|---|---|---|---|---|---|---|
| **Data collection and processing** | | | | | | |
| Magnification | 105,000 | | | 105,000 | | |
| Voltage (kV) | 300 | | | 300 | | |
| Electron exposure (e–/Å$^2$) | ~42 e$^-$ per Å$^2$ | | | ~40 e$^-$ per Å$^2$ | | |
| Defocus range (μm) | −0.7 to −1.7 μm | | | −0.7 to −1.7 μm | | |
| Pixel size (Å) | 0.837 | | | 0.837 | | |
| Symmetry imposed | $C_1$ | | | $C_1$ | | |
| Initial particle images (no.) | 2,494,826 | | | 2,088,573 | | |
| Final particle images (no.) | 38,385 | | | | | |
| Map name | NDH–PSI–LHCI-2 supercomplex (EMD-19244) | NDH peripheral arm (EMD-19241) | NDH membrane arm (EMD-19246) | NDH–PSI–LHCI-2b order (EMD-19247) | PSI–LHCI-2 (EMD-19248) | Composite map (EMD-51527) |
| Description | Global refined map | Local refined map | Local refined map | Local refined map | Local refined map | Composite map |
| FSC threshold | 0.143 | 0.143 | 0.143 | 0.143 | 0.143 | / |
| Map resolution (Å) | 3.34 | 3.27 | 3.17 | 3.26 | 3.09 | 3.2 (average) |
| Map resolution range (Å) | 3.23–8.12 | 3.22–48.65 | 3.17–5.17 | 3.26–5.07 | 3.04–4.95 | / |
| **Refinement** | | | | | | |
| Initial model used (PDB code) | / | 6KHJ and AlphaFold[33] | 6KHJ and AlphaFold[33] | 6KHJ and AlphaFold[33] | 4Y28 (PSI–LHCI) | Models from local refined maps |
| Map sharpening B factor (Å$^2$) | 82.4 | 87 | 91 | 93.1 | 84.7 | / |
| | Final model and map used for validation: PDB 9GRX and composite map EMD-51527 | | | | | |
| Model resolution (Å) FSC threshold (0.5) | 3.3 | | | | | |
| Model resolution average (Å) | 3.2 | | | | | |
| Model resolution range (Å)$^a$ | 0.01–2.78 | | | | | |
| Final model composition | | | | | | |
| Nonhydrogen atoms | 83,867 | | | | | |
| Protein residues | 9,086 | | | | | |
| Ligands | 248 | | | | | |
| *B* factors (Å$^2$, min/max/mean) | | | | | | |
| Protein | 15.69/186.07/55.65 | | | | | |
| Ligand | 9.93/218.66/66.00 | | | | | |
| Root-mean-square deviations (r.m.s.d.) | | | | | | |
| Bond lengths (Å) | 0.007 | | | | | |
| Bond angles (°) | 1.191 | | | | | |
| Validation | | | | | | |
| MolProbity score | 1.94 | | | | | |
| Clashscore | 8.7 | | | | | |
| Poor rotamers (%) | 2.22 | | | | | |
| Ramachandran plot | | | | | | |
| Favored (%) | 96.69 | | | | | |
| Allowed (%) | 3.15 | | | | | |
| Disallowed (%) | 0.17 | | | | | |
| Ramachandran plot *Z*-score (r.m.s.d.) | | | | | | |
| Whole (*n*=8990) | 0.06 (0.09) | | | | | |
| Helix (*n*=4704) | 0.67 (0.07) | | | | | |
| Sheet (*n*=686) | 0.03 (0.20) | | | | | |
| Loop (*n*=3600) | -0.70 (0.10) | | | | | |

$^a$The model resolution range (Å) is reported as the min/max local resolution values measured at model atom positions. It was calculated using the 'values at atom positions' tool in Chimera.

## Data availability

The following maps were deposited to the EM Data Bank: EMD-51527 (composite map, NDH–PSI–LHCI-2 supercomplex), EMD-19244 (complete map, NDH–PSI–LHCI-2 supercomplex), EMD-19241 (local refined peripheral arm of NDH), EMD-19246 (local refined membrane arm of NDH), EMD-19247 (local refined border region between NDH and PSI–LHCI-2) and EMD-19248 (local refined PSI–LHCI-2). The atomic model of the NDH–PSI–LHCI-2 supercomplex was deposited to the PDB under accession code 9GRX.

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

## Acknowledgements

We thank J. Vonck and Ö. Yildiz for the useful discussions. We thank the Central EM Facility of the Max Planck Institute of Biophysics and S. Welsch for support in the acquisition of the single-particle cryo-EM data. We thank J. F. Castillo-Hernandez and Ö. Yildiz for information technology support. Funding was obtained from the Max Planck Society.

## Author contributions

A.H. and W.K. initiated the project. A.H. purified the supercomplex from *S. oleracea* and collected cryo-EM data. B.I. performed image processing, built and analyzed the atomic models and produced the figures. All authors evaluated the data. B.I. and W.K. wrote the manuscript, with contributions from A.H.

## Funding

## Competing interests

The authors declare no competing interests.

## Additional information

**Extended data** is available for this paper at https://doi.org/10.1038/s41594-024-01478-1.

**Correspondence and requests for materials** should be addressed to Werner Kühlbrandt.

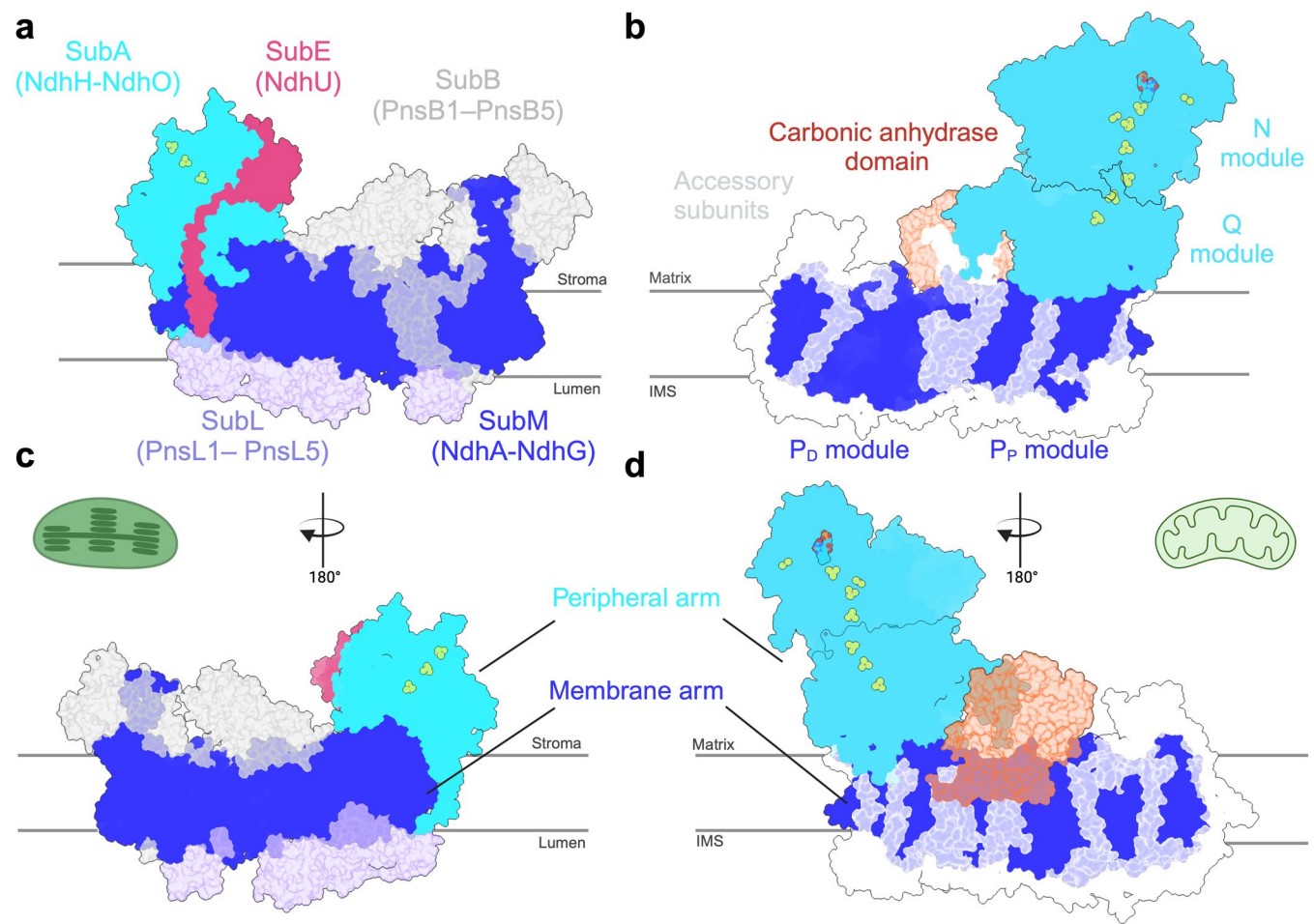

**Extended Data Fig. 1 | NDH is the chloroplast homolog of complex I.** Lateral views of (**a**) and (**c**) *S. oleracea* NDH (PDB ID: 9GRX) and (**b**) and (**d**) *A. thaliana* complex I (PDB IDs: 8BEF, 8BEH, 8BED). The different modules forming those complexes are color coded. Conserved modules are shown in the same color in both models.

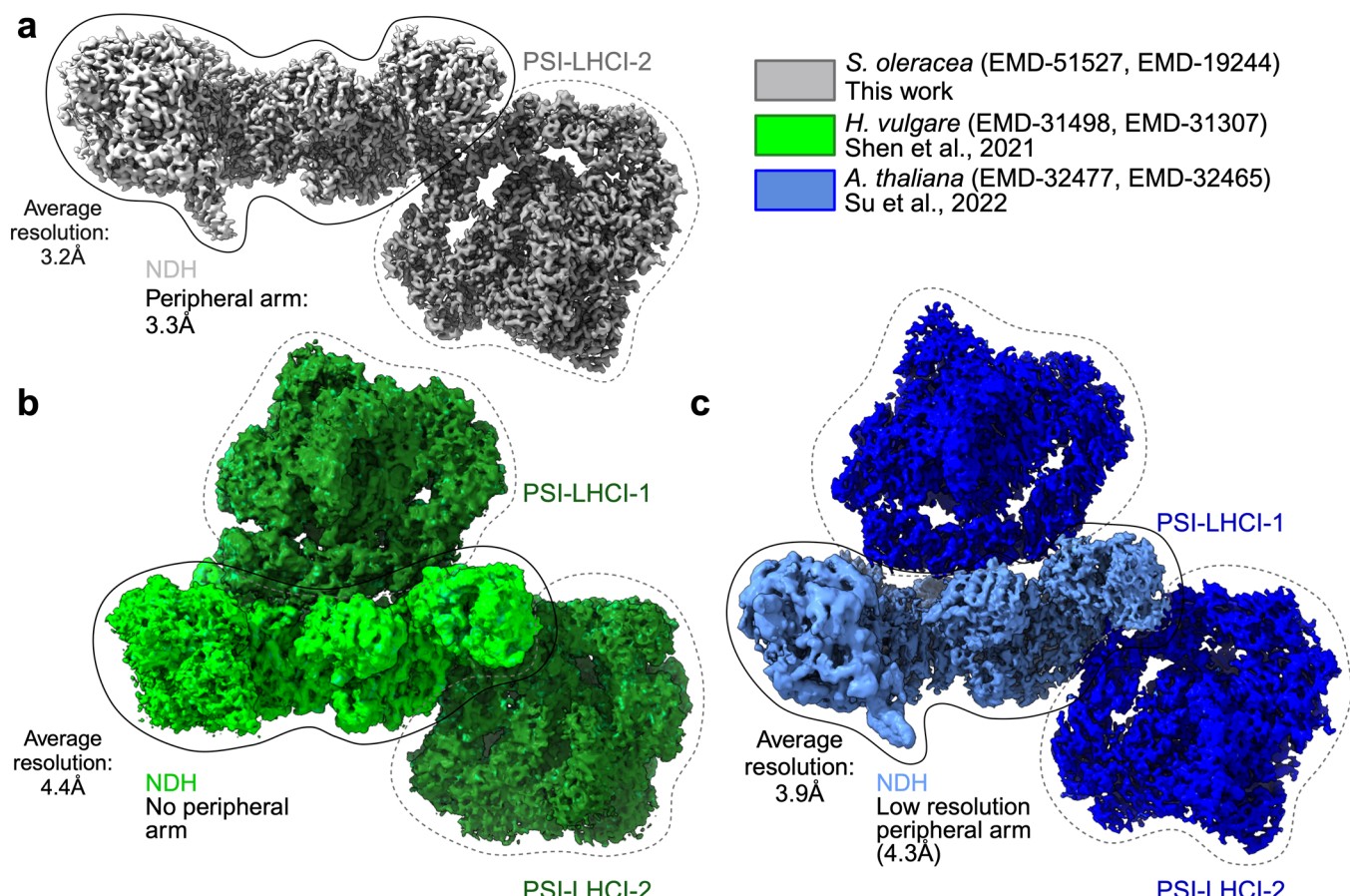

**Extended Data Fig. 2 | Spinach NDH-PSI-LHCI-2 supercomplex contains one copy of PSI.** Top views of the cryoEM maps of NDH-PSI-LHCIs supercomplexes from (**a**) *S. oleracea* (contour level 0.38), (**b**) *H. vulgare* (contour level 5.5) and (**c**) *A. thaliana* (contour level 0.023). Average resolutions and main elements forming the complexes are indicated.

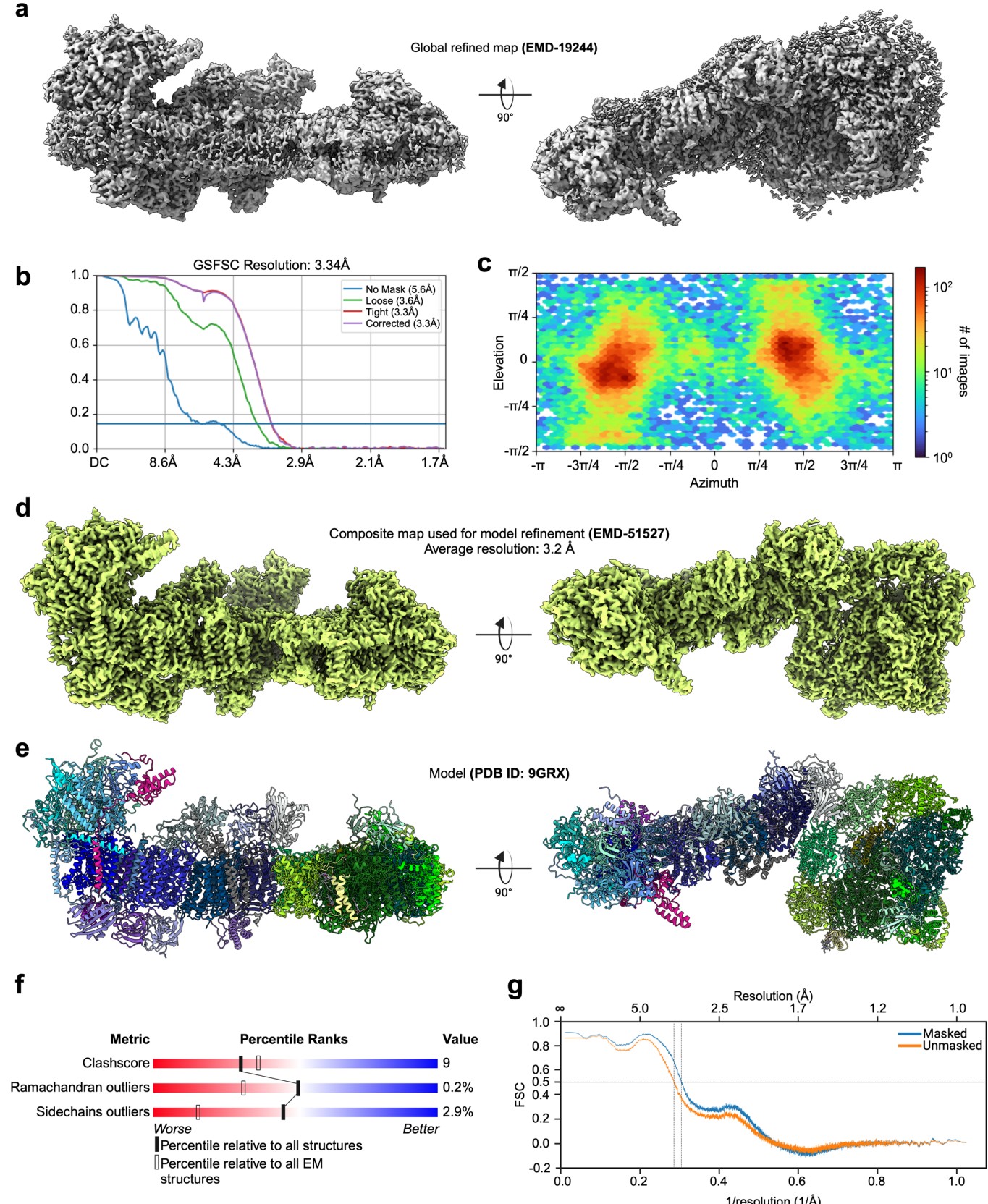

**Extended Data Fig. 3 | See next page for caption.**

**Extended Data Fig. 3 | NDH-PSI-LHCI-2 full maps and model. a** and **b**, Globally refined cryoEM map (contour level of 0.15) of the supercomplex from *S. oleracea* with FSC curve (**b**). This map was used for local refinement as shown in Extended Data Fig. 4. **c**, Angular distribution calculated in cryoSPARC[63] for the 38,385 particles included in this final reconstruction. The Heat map (**c**) indicates the number of particles for each viewing angle. **d**, Composite map (contour level 0.38) created by merging the four locally refined maps showed in Extended Data

Fig. 4 and e, model of the NDH-PSI-LHCI-2 supercomplex. EMDB and PDB IDs are indicated. **f**, Validation slider shows the quality assessment of the model (PDB ID: 9GRX). with bars representing the clashscore, Ramachandran outliers and Sidechain outliers for the structure. **g**, The composite map was used as consensus for the refinement and validation of the model. Map-versus-model FSC curves with and without mask were calculated using Mtriage in PHENIX[68].

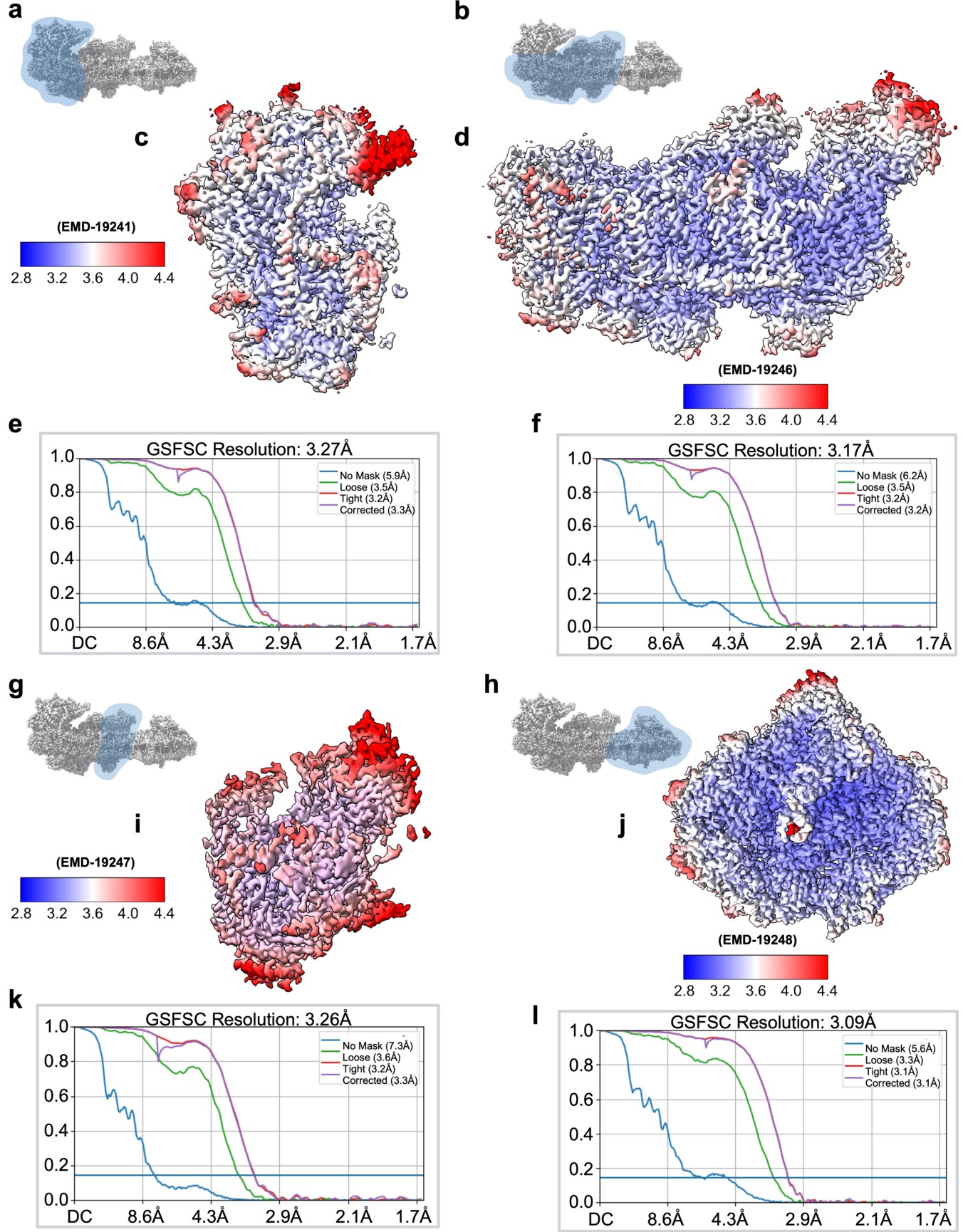

**Extended Data Fig. 4 | Locally refined maps of NDH-PSI-LHCI-2. a, b, g, h**, Side view of the supercomplex map (EMD-19244, Extended Data Fig. 3a) shown as grey surface. Regions used for each local refinement are outlined in blue. **c, d, i, j**, Side views of different locally refined cryoEM maps colored by local resolution with FSC curves in **e, f, k, l**. The EMBD IDs for each map are indicated above the color key. CryoEM density maps are drawn at contour levels 0.3 (**c**), 0.4 (**d**), 0.3 (**i**) and 0.2 (**j**).

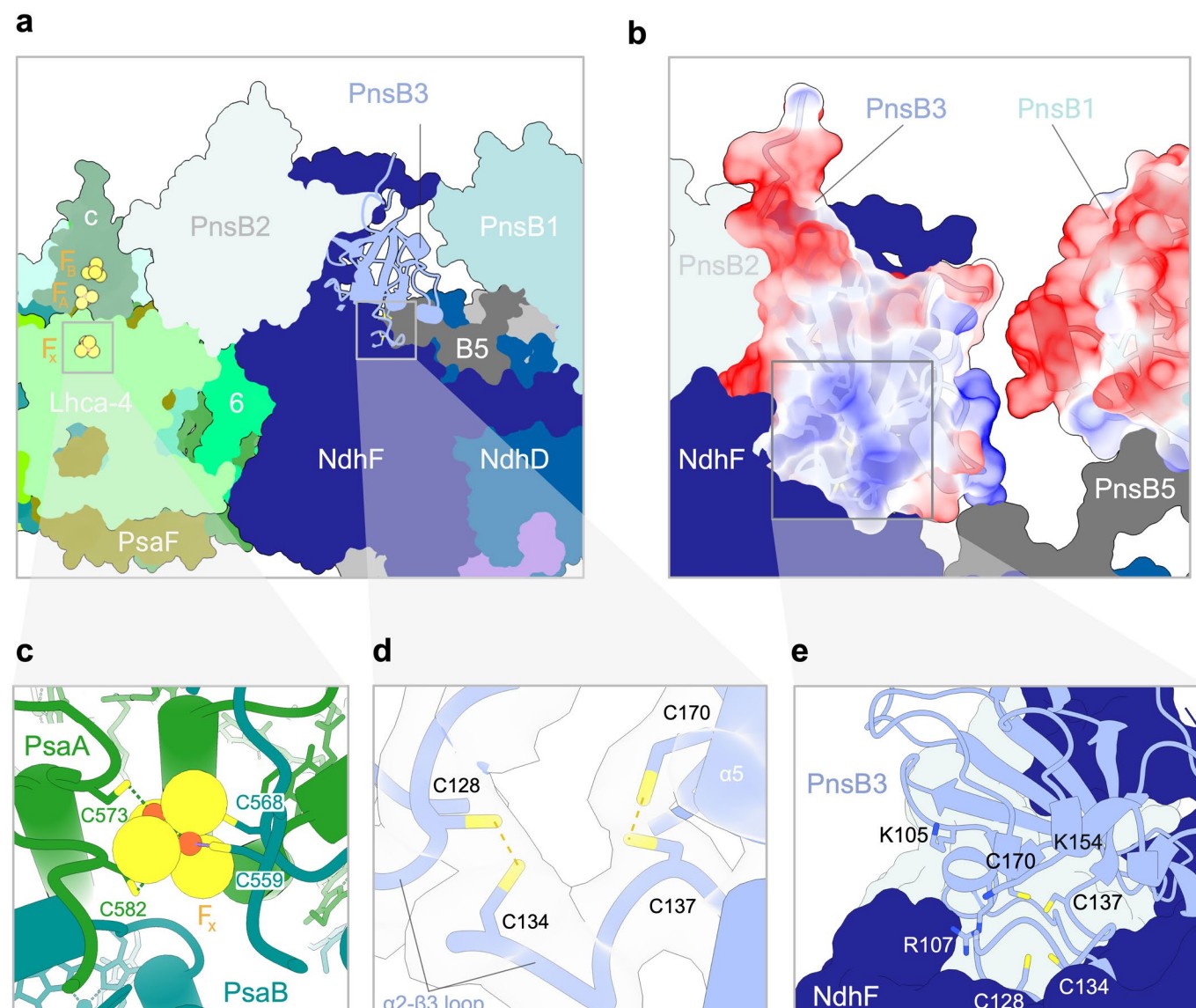

**Extended Data Fig. 5 | Fe-S clusters. a**, Lateral view of the supercomplex. PnsB3 is shown as cylinders and the 4Fe-4S clusters in PSI as spheres. Other subunits are shown in outline. The highlighted regions are illustrated in C and D. **b**, PnsB1 and PnsB3 shown as electrostatic potential surfaces. Red, negative charge; blue, positive charge. Cartoons of both PnsB1 and PnsB3 can be seen through the surface. Other subunits are shown in outline. The highlighted region (illustrated in **e**) shows the positively charged surface on PnsB3 that may bind Fd. **c**, Cysteins from PsaA and PsaB that contribute to the formation of the first 4FE-4S cluster ($F_x$) in the PSI reaction center. **d**, Cysteine residues within PnsB3 predicted to form a Fe-S cluster. The cryoEM map is shown (contour level 0.25) and the cysteine residues are labelled. Density for the Fe-S cluster is not present. **e**, Positively charged residues of PnsB3 that potentially interact with negatively charged Fd residues are shown as sticks and are labelled. The cysteines cluster is close to this positive latch (less than 10 Å).

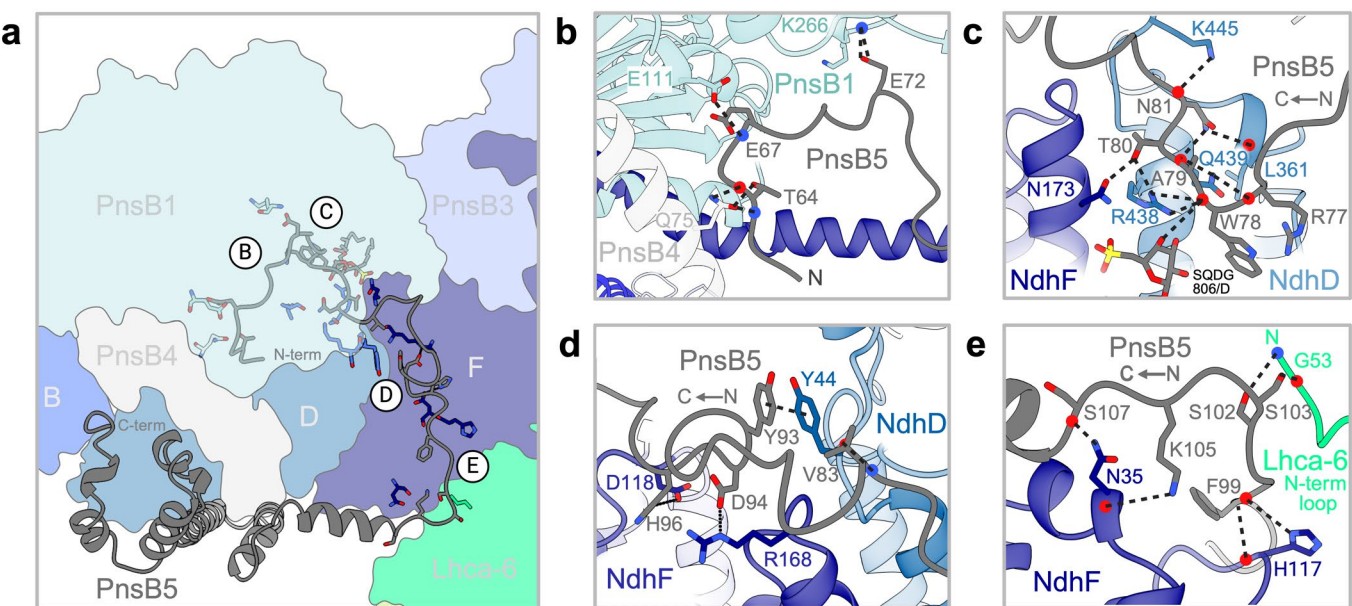

**Extended Data Fig. 6 | PnsB5 N-terminal loop contacts different NDH subunits. a**, Top view of Pnsb5 N-terminal loop. Key residues involved in electrostatic interactions are shown as sticks and are labelled with numbers (B-E) corresponding to panels **b-e**. The neighboring subunits are shown as surfaces. **b-e**, Details of the interactions between PnsB5 N-terminal loop and nearby subunits.

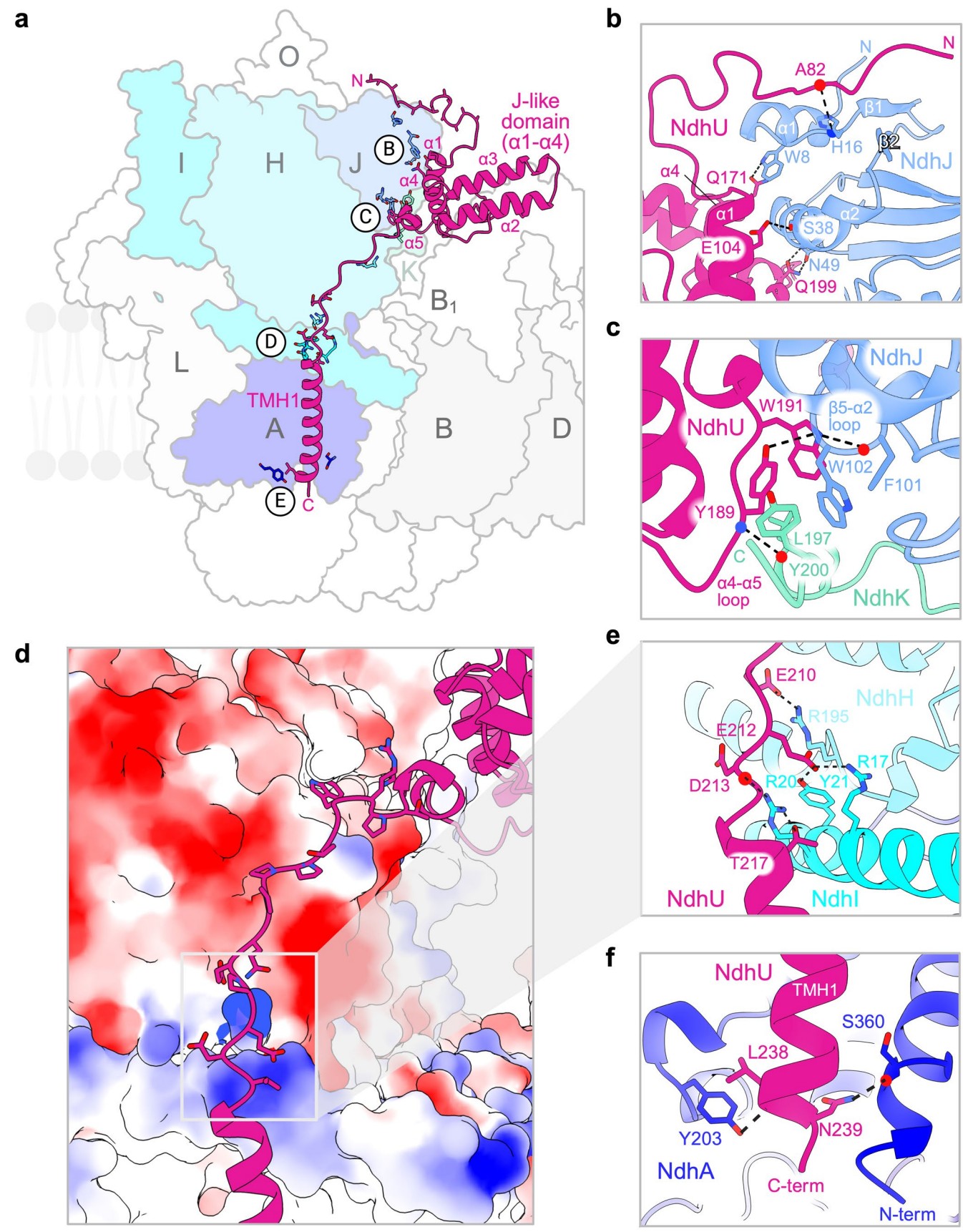

**Extended Data Fig. 7 | See next page for caption.**

**Extended Data Fig. 7 | NdhU interacts directly with modules SubA and SubM.**
**a**, Lateral view of the peripheral arm. NdhU is shown as a ribbon diagram in magenta. Subunits NdhA, NdhH, NdhI, NdhJ and NdhK interacting with NdhU are shown in outline and are color-coded as in Fig. 1. Residues forming hydrogen bonds are indicated by letters (**b**) to (**e**). **d**, Surface view of the peripheral arm with the loop connecting the J-like domain to the transmembrane helix shown in cartoon representation. Interacting subunits are shown as electrostatic potential surfaces (red, negative; blue, positive). **b, c, e, f**, Atomic subunit interactions. Letters refer to panel (**a**).

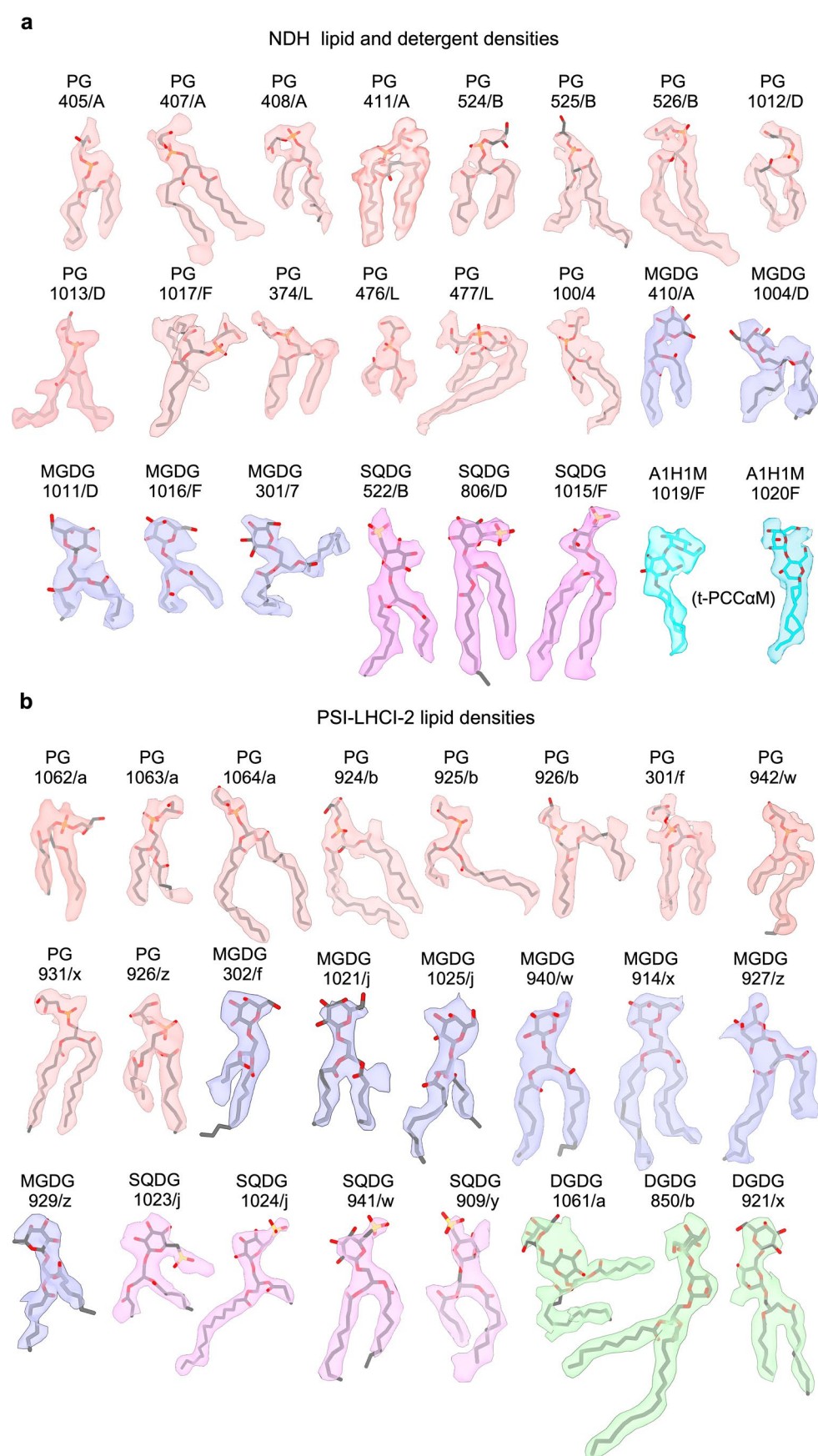

**a** NDH lipid and detergent densities

**b** PSI-LHCI-2 lipid densities

**Extended Data Fig. 8 | NDH-PSI-LHCI-2 supercomplex contains at least 46 well-defined lipids and two detergent molecules.** Overview of observed lipid densities in (**a**) NDH and (**b**) PSI-LHCI-2. Lipid models are fitted into the respective cryoEM density (contour level 0.2–0.4) that is color-coded by lipid identity: PG, salmon; MGDG, violet; SQDG, pink; DGDG, green. **c**, CryoEM density of the t-PCCαM detergent (three letter code: A1H1M) in turquoise. Each lipid is identified by its residue number, followed by a backslash and the chain identifier.

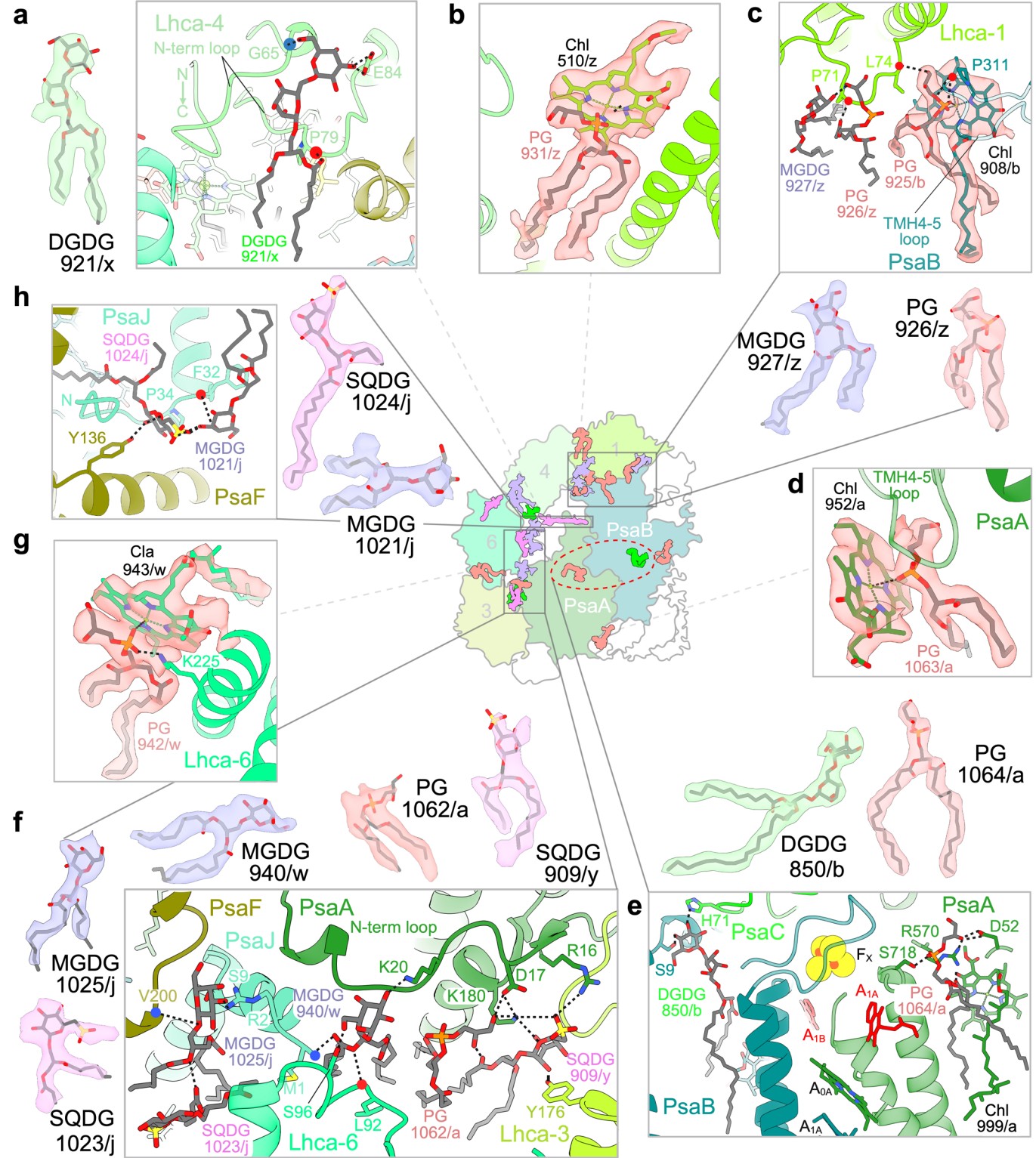

**Extended Data Fig. 9 | Lipids in the PSI-LHCI-2 supercomplex.** The central panel shows the outlines of the membrane-spanning subunits in the PSI-LHCI-2 complex that interact with lipids. PsaA and PsaB of the PSI core are color-coded. Light-harvesting complexes Lhca-3, Lhca-6, Lhca-4 and Lhca-1 are labelled 3,4,6, and 1, respectively. Stromal subunits have been removed for clarity. Lipids (contour level 0.2-0.4) are color-coded as follows: phosphatidyl glycerol (PG), salmon; monogalactosyl diacylglycerol (MGDG), violet; sulfoquinovosyl diacylglycerol (SQDG), pink; digalactosyl diacylglycerol (DGDG), green. **a-h,**

Details of lipid-protein and lipid-chlorophyl interactions. **a, c, e, f,** and **h,** Lipids at the PSI-LHCI-2 interface create hydrophobic bridges between neighboring subunits. **b-d,** and **g,** PG often closely interacts with chlorophylls. The cryoEM density is shown in salmon at contour levels 0.2-0.3. **e,** Details of interactions established by PG 1064/a (salmon density) and DGDG 850/b (green density) close to the reaction center. The two lipids are highlighted by a broken red line in the central panel.

**a**

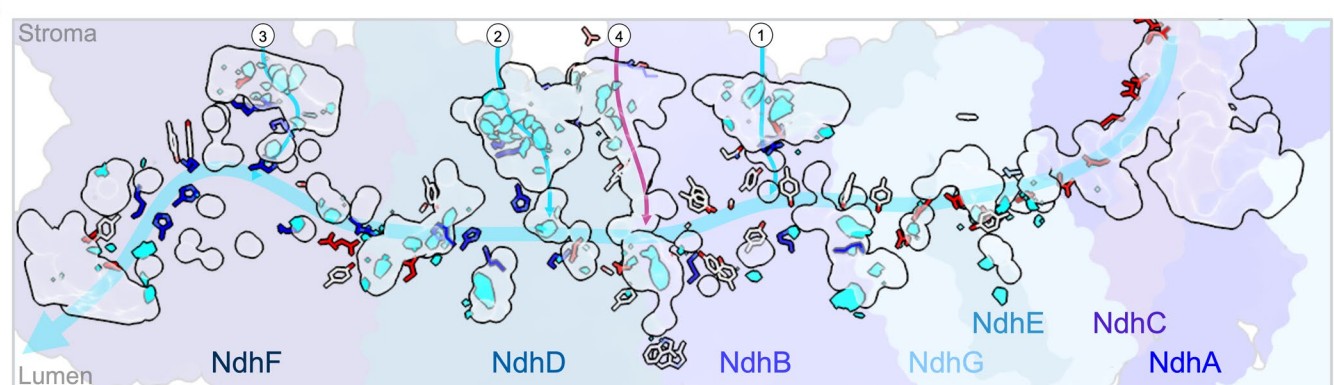

**Extended Data Fig. 10 | Potential water densities in the voids, cavities and channels of the NDH membrane arm. a**, Side view of NDH. The core transmembrane subunits are depicted as surfaces, color-coded as in Fig. 1. The outlines in black represent the map of voids, channels, and depressions within NDH generated with HOLLOW software tool[35], that are believed to play a crucial role in proton translocation. Charged residues of NdhA-G and residues involved in proton translocation that are conserved between spinach NDH and complex I are shown as stick models and are color-coded. Positively charged residues are blue, negatively charged residues are red and other residues are white. The continuous aqueous passage, defined by hydrophilic residues from the Q-binding site to NdhF, is indicated by the light-blue arrow. Stromal half-channels positioned above NdhB, NdhD, and NdhF are indicated with turquoise arrows and labelled from 1 to 3. The channel between NdhB and D, referred to as the NdhB-D channel, is marked with a magenta arrow and labelled as number 4. Unresolved map features within the cavities are cyan.

# Reporting Summary

## Statistics

For all statistical analyses, confirm that the following items are present in the figure legend, table legend, main text, or Methods section.

| n/a | Confirmed | |
|---|---|---|
| ☐ | ☒ | The exact sample size (*n*) for each experimental group/condition, given as a discrete number and unit of measurement |
| ☐ | ☒ | A statement on whether measurements were taken from distinct samples or whether the same sample was measured repeatedly |
| ☒ | ☐ | The statistical test(s) used AND whether they are one- or two-sided<br>*Only common tests should be described solely by name; describe more complex techniques in the Methods section.* |
| ☒ | ☐ | A description of all covariates tested |
| ☒ | ☐ | A description of any assumptions or corrections, such as tests of normality and adjustment for multiple comparisons |
| ☒ | ☐ | A full description of the statistical parameters including central tendency (e.g. means) or other basic estimates (e.g. regression coefficient) AND variation (e.g. standard deviation) or associated estimates of uncertainty (e.g. confidence intervals) |
| ☒ | ☐ | For null hypothesis testing, the test statistic (e.g. *F*, *t*, *r*) with confidence intervals, effect sizes, degrees of freedom and *P* value noted<br>*Give P values as exact values whenever suitable.* |
| ☒ | ☐ | For Bayesian analysis, information on the choice of priors and Markov chain Monte Carlo settings |
| ☒ | ☐ | For hierarchical and complex designs, identification of the appropriate level for tests and full reporting of outcomes |
| ☒ | ☐ | Estimates of effect sizes (e.g. Cohen's *d*, Pearson's *r*), indicating how they were calculated |

*Our web collection on statistics for biologists contains articles on many of the points above.*

## Software and code

Policy information about availability of computer code

| | |
|---|---|
| Data collection | EPU 2.8.1, EPU 2.9.0 |
| Data analysis | cryoSPARC v3.2 - cryoSPARC v4.2.1, Topaz 0.2.5a, RELION 3.0, COOT 8.0, UCSF ChimeraX 1.6, Isolde, Phenix 1.21, AlphaFold2, Molprobity 4.5.2 |

For manuscripts utilizing custom algorithms or software that are central to the research but not yet described in published literature, software must be made available to editors and reviewers. We strongly encourage code deposition in a community repository (e.g. GitHub). See the Nature Portfolio guidelines for submitting code & software for further information.

## Data

Policy information about availability of data

All manuscripts must include a data availability statement. This statement should provide the following information, where applicable:

- Accession codes, unique identifiers, or web links for publicly available datasets
- A description of any restrictions on data availability
- For clinical datasets or third party data, please ensure that the statement adheres to our policy

The maps have been deposited to EMDB: EMD-51527 (composite map, NDH-PSI-LHCI-2 supercomplex); EMD-19244 (complete map, NDH-PSI-LHCI-2 supercomplex); EMD-19241 (local refined peripheral arm of NDH); EMD-19246 (local refined membrane arm of NDH); EMD-19247 (local refined border region between NDH and PSI-LHCI-2); EMD-19248 (local refined PSI-LHCI-2). The atomic model of NDH-PSI-LHCI-2 supercomplex has been deposited with the PDB ID: 9GRX.
PDB ID: 4Y28 and PDB ID: 6KHJ were used as consensus structures for the initial construction of the here presented model.

# Research involving human participants, their data, or biological material

Policy information about studies with [human participants or human data](). See also policy information about [sex, gender (identity/presentation), and sexual orientation]() and [race, ethnicity and racism]().

| | |
|---|---|
| Reporting on sex and gender | N/A |
| Reporting on race, ethnicity, or other socially relevant groupings | N/A |
| Population characteristics | N/A |
| Recruitment | N/A |
| Ethics oversight | N/A |

Note that full information on the approval of the study protocol must also be provided in the manuscript.

# Field-specific reporting

Please select the one below that is the best fit for your research. If you are not sure, read the appropriate sections before making your selection.

☒ Life sciences  ☐ Behavioural & social sciences  ☐ Ecological, evolutionary & environmental sciences

For a reference copy of the document with all sections, see [nature.com/documents/nr-reporting-summary-flat.pdf](http://nature.com/documents/nr-reporting-summary-flat.pdf)

# Life sciences study design

All studies must disclose on these points even when the disclosure is negative.

| | |
|---|---|
| Sample size | No statistical method was used to predetermine the sample size. The number of micrographs collected was determined by the observed particle distribution per image at the necessary magnification (pixel size) for reconstructing density maps at near-atomic resolution, which is essential for unbiased model building and interpretation. Following dataset classification, the remaining particles were sufficient for reconstructing a density map of the NDH-PSI-LHCI-2 supercomplex at a resolution of 3.2 Å. Additionally, local density maps were generated at resolutions of 3.3 Å, 3.2 Å, and 3.1 Å for the NDH peripheral arm, NDH membrane arm, the interface between NDH and PSI-LHCI-2, respectively. |
| Data exclusions | No data was excluded from initial data processing. Exclusion was later performed by unbiased 2D and 3D classification as part of the described data processing pipelines. Particle classes that form features-enriched 2D class averages or 3D back projections were manually selected, combined and used for further rounds of 2D/3D classification and selection. Additional details can be found in Supplementary Figures 12 and 13, as well as in the Methods section.. |
| Replication | Not applicable. Replication of entire datasets for structural biology studies is not applicable due to technical limitations. However, reconstructions of 3D maps represent an weighted average of thousands of individual particle images contained within the final dataset. |
| Randomization | We utilized the Gold standard Fourier Shell Correlation method to assess the resolution of the cryo-EM structures. This method involves splitting the dataset into two sets, odd and even, which are then refined independently. The splitting of the dataset is random. |
| Blinding | Not applicable. |

# Reporting for specific materials, systems and methods

We require information from authors about some types of materials, experimental systems and methods used in many studies. Here, indicate whether each material, system or method listed is relevant to your study. If you are not sure if a list item applies to your research, read the appropriate section before selecting a response.

## Materials & experimental systems

| n/a | Involved in the study |
|---|---|
| ☒ | Antibodies |
| ☒ | Eukaryotic cell lines |
| ☒ | Palaeontology and archaeology |
| ☒ | Animals and other organisms |
| ☒ | Clinical data |
| ☒ | Dual use research of concern |
| ☐ | ☒ Plants |

## Methods

| n/a | Involved in the study |
|---|---|
| ☒ | ChIP-seq |
| ☒ | Flow cytometry |
| ☒ | MRI-based neuroimaging |

# Plants

| | |
|---|---|
| Seed stocks | In this study, we opted not to utilize seed stock as our primary source material. Instead, we used young leaves obtained directly from market spinach (Spinacia oleracea). |
| Novel plant genotypes | N/A |
| Authentication | N/A |

