## [Peer Review File · Nature Structural & Molecular Biology]

CryoEM structure of the NDH-PSI-LHCI supercomplex from *Spinacia oleracea*

Corresponding Author: Professor Werner Kühlbrandt

Version 0:

Decision Letter:

9th May 2024

Dear Dr. Kühlbrandt,

Thank you again for submitting your manuscript "CryoEM structure of the NDH-PSI-LHCI-2 supercomplex from *Spinacia oleracea*". We now have comments (below) from the 3 reviewers who evaluated your paper. In light of those reports, we remain interested in your study and would like to see your response to the comments of the referees, in the form of a revised manuscript.

You will see that while reviewers appreciate the results, they raise several concerns which will need to be addressed in a revision. Specifically, please address the concerns of reviewer #1 regarding the model building. In line with their further comments, please comment on the lack of iron-sulfur cluster in the Pnsb3 subunit. Please also ensure to respond to referee #3 comments regarding the complex naming conventions, as well as all points referring to the particulars of cryo-EM data collection and analysis, as well as consider the points for discussion brought up by referee #2.

Please be sure to address/respond to all concerns of the referees in full in a point-by-point response and highlight all changes in the revised manuscript text file. If you have comments that are intended for editors only, please include those in a separate cover letter.

We expect to see your revised manuscript within 12 weeks. If you cannot send it within this time, please contact us to discuss an extension; we would still consider your revision, provided that no similar work has been accepted for publication at NSMB or published elsewhere.

Reporting Summary:

Please note that all key data shown in the main figures as cropped gels or blots should be presented in uncropped form, with molecular weight markers. These data can be aggregated into a single supplementary figure item. While these data can be displayed in a relatively informal style, they must refer back to the relevant figures. These data should be submitted with the final revision, as source data, prior to acceptance, but you may want to start putting it together at this point.

Data availability: this journal strongly supports public availability of data. All data used in accepted papers should be available via a public data repository, or alternatively, as Supplementary Information. If data can only be shared on request, please explain why in your Data Availability Statement, and also in the correspondence with your editor. Please note that for some data types, deposition in a public repository is mandatory - more information on our data deposition policies and available repositories can be found below:

<https://www.nature.com/nature-research/editorial-policies/reporting-standards#availability-of-data>

Link Redacted

Sincerely,

Katarzyna Ciazynska, PhD
(she/her)
Associate Editor
Nature Structural & Molecular Biology
<https://orcid.org/0000-0002-9899-2428>

Referee expertise:

Referee #1: photosynthesis, cryo-EM

Referee #2: respiratory supercomplexes

Referee #3: respiratory supercomplexes, cryo-EM

Reviewers' Comments:

Reviewer #1:

Remarks to the Author:

REVIEW OF NSMB-A49056

General comments:

The authors present the cryo-EM structure of the NDH-PSI-LHCI-2 supercomplex from spinach at a global resolution of 3.2 Å. It is a third recent work describing a similar NDH-PSI-LHC supercomplex from vascular plants. Here, authors manage to obtain a higher overall resolution of the supercomplex than previously reported, providing insights into subunit interactions and cofactors unresolved in other organisms. The work is a purely structural study. Results are exhaustive and clearly presented overall. Points that need further clarification are outlined below. The comparative approach, both with homologous models of NDH-PSI-LHCI complexes and mitochondrial Complex I is much appreciated.

Novel biological insight focuses on a more precise description of the NDH complex, including assignment of the NdhU subunit and the first elucidation of the full PnsB1 subunit and its interactions within the supercomplex. However, the discussion of the potential missing iron-sulfur cluster in the Pnsb3 subunit is lackluster. Can it be the effect of the different purification protocol or have a physiological meaning? Could authors comment on that? Second, from the overall calculation, chlorophyll and carotenoids content per PSI-LHCI of the NDH-PSI-LHCI-2 reported here seems greater (154 and 38 versus (includes two PSI-LHCI) 298 and 67 respectively) than that in the NDH-PSI-LHCI-1 supercomplex from barley. Previous studies reported different Chl content of Lhca5 and Lhca6 subunits compared to other LHCI and counts varied between species. This should be clarified in the text.

While the cryo-EM imaging and data processing methods are solid (pending only minor clarifications, see below), the modeling of the atomic structure needs some reevaluation. As evidenced by the structural validation report, the overall model quality does not seem to be on par with the map quality. In particular, the clash score and the sidechain outliers stick out. One cycle (or a few) of real-space refinement in PHENIX should be able to fix most issues automatically at this stage. Care should be taken in particular with the nonbonded_weight parameter (the default value of 100 in PHENIX is too low for typical cryo-EM maps, especially of large complexes) and the rotamer fitting procedures (e.g. local grid search should help). The authors should nevertheless carefully reexamine the model and its fit to the experimental density afterwards, to evaluate any significant changes that could affect their conclusions, and remake figures when necessary.

Specific comments:

L390 Image processing for single particle cryo-EM

The authors should explain their rationale for importing their project from cryoSPARC into RELION for specific refinement tasks (3D refine, CTF refine, Bayesian polishing) and back into cryoSPARC.

L416 Model building and analysis

Figure 1:

The current choices of color and lighting make it difficult to understand the architecture and details of this intricate cryo-EM map. We suggest the authors experiment with ambient occlusion lighting (for example as implemented in ChimeraX) and surface outlines to improve interpretability. Furthermore, the labels indicating the different proteins and domains are difficult to read, a font outline could be helpful here as well.

Figure 4G:

The densities for Y222 and F232 are not well resolved in comparison to other side chains in the same helix, at least at the isosurface contour shown; could the authors comment on it? For example, are there nearby interactions (or lack thereof) affecting the resolution of these densities?
(See general comment about model quality above).

Supp. Fig 2:

The FSC plots should be bigger; they are difficult to evaluate as they are currently.

While we acknowledge there is no defined standard in the field for how local-resolution maps should be presented, we suggest that reversing the color-coding would be more intuitive: red for lower resolution parts (more disordered, "hot") and blue for the highest-resolution parts (well-ordered, "cold") of the map. This is not essential and should only be done at the authors' discretion.

The model-map FSC curve should also be presented, in this figure or elsewhere.

Supp. Fig. 19:

This figure should be "closer" to Supp. Fig. 2 (either before or after it) as they are directly related.

The figure panels should be rearranged so that the example micrographs and 2D classes can be shown bigger; it is quite difficult to evaluate them currently.

A common issue in single-particle cryo-EM, in particular for membrane protein complexes, is preferential orientation. The authors should include an angular distribution plot to aid interpretation of their experimental density.

Supp. Table 2:

The model-map resolution from the FSC should be evaluated at the FSC=0.5 threshold, as it is comparing a noiseless structural model to a noisy map (see Rosenthal & Henderson 2003 and Phenix validation documentation for details).

Minor comments:

The Discussion section is basically a repetition of the results and conclusions drawn by authors and others; especially paragraphs comprising L297-L318 and L330-L339.

E.g.: L297-L298: "Our structure shows that Lhca-6 interacts with the SubB module of NDH, confirming earlier speculations" – That interaction was quite firmly confirmed before.

In our opinion it would be more interesting for the reader to speculate on differences between NDH-PSI-LHCI-1 and -2 supercomplexes reported in the literature. The authors are rare experts in the physiological-structural studies; their comments on points such as the importance of two-point stabilization of the NDH complex by the PSI-LHC1s or potential heterogeneity of the supercomplexes in vivo could be particularly valuable.

Typos and others:

We have been recently informed by one of the Nature Publishing Group editors to avoid the term "higher plant". We propagate this information to the authors.

"Lhca-6" appears in the text also as "Lhca6" - is that intentional?

L183: crystal structure of in *P. sativum* PSI-LHCI

L377: screed (screened)

L728: Figure 4: Identification NdhU (Identification of NdhU)

Reviewed by:

Ricardo D. Righetto and Wojciech Wietrzynski

Reviewer #2:

Remarks to the Author:

In their manuscript "CryoEM structure of the NDH-PSI-LHCI-2 supercomplex from *Spinacia oleracea*" Introini et al. present the structure of the titular photosynthetic supercomplex at significantly improved resolution compared with previous work. This allows the author to build models at near atomic resolution for the entire supercomplex and identify additional subunits that were lacking in previous reconstructions and models. Additionally, the authors are able to use their improved models to compare and contrast the structural features of the chloroplast NDH complex to structures from cyanobacteria and respiratory complex I. The approach and data are valid and of high quality. Altogether the paper is very well written and will be of wide interest to those in the field of bioenergetics, plant physiology and membrane protein structural biology.

Strengths:

- 1) improved structure of the NDH-PSI-LHCI-2 supercomplex allow for the modeling of previously low-resolution domains.
- 2) detailed characterization of interactions between NDH and PSI-LHCI-2 complexes.
- 3) identification of the J-like NdhU subunit.
- 4) identification of several lipid binding sites that are conserved in homologous complexes.
- 5) description of the aqueous half channels and hydrophilic axis in the membrane domain.
- 6) comparison of NDH to respiratory complex I.

The paper could be improved by a discussion in the intro about the relative abundance of the NDH complex. Relative to linear photosynthesis how much flux is known to occur through the cyclic electron flow (CEF) pathway and is the abundance of the NDH complexes in thylakoids sufficient to accommodate the levels of flux observed. There is controversy in the literature regarding the degree to which NDH contributes to CEF (see pmid: 30827891 for example) as it is relatively low abundance (1:100 ratio of NDH:PSI in tobacco leaves; pmid: 9463365) and low flux (2.5 e-/s*PSI compared to CEF flux of 60 e-/s*PSI measured in *Chlamydomonas*; pmid: 30498023, 30827891) . How can the structure presented here inform on this controversy.

Minor points:

Page 2, line 50: the authors write "NDH takes the role of the mitochondrial NADH:ubiquinone oxidoreductase (complex I)", however, given that the reaction it catalyzes is different this language is too strong, perhaps change to "NDH is analogous to..."

Page 2, line 58: the authors refer to the ubiquinone reduction (Q) domain of complex I" it is more common to refer to this

region of complex I as the “Q module” as opposed to domain and the authors do so later in the text, good to be consistent throughout.

Page 3, line 87: the authors state that “We detected 26 out of the 29 NDH subunits...” but do not indicate how these were “detected” are they referring to what they are able to model in the reconstruction or detect on a gel or by mass spec?

Page 4, line 130: the author state “A detergent molecule replacing a lipid...” but do not state how they determined that this site would normally be occupied by a lipid in thylakoids, given that no lipid is observed in their structure at this position perhaps this phrase should be made more speculative, i.e., “likely replacing a lipid...”

Supplementary figure 6 is not referred to in the text.

Page 6, line 191: the authors refer to “the PQ entry channel, E channel and PQ binding pocket of complex I” however, complex I utilizes ubiquinone not plastoquinone.

Supplementary figure 1: “Those that are conserved are...” is a bit ambiguous, perhaps rewrite “Conserved modules are...”

Supplementary figure 2: The FSC plots are too small and need to be made larger with larger font.

Supplementary figure 9, line 959: the authors refer to “the continuous aqueous passage...” however according to the figure the aqueous passage is not continuous, perhaps “continuous hydrophilic passage...” is what was meant?

Reviewer #3:

Remarks to the Author:

The authors present the *Spinacia oleracea* NDH-PSI-LHCI supercomplex at an improved resolution relative to previous publications on this supercomplex (and the related NDH-PSI2-LHCI) of different species, in part due to using focus refinements on sub-regions. The authors used focused classification to better resolve the peripheral domain of the NDH complex, resulting in improved definition of elements that compose it and the identification of the NdhU subunit at this site also. The improved resolution of NDH allowed for the modelling of the N-terminal region of PnsB1, and a comparison to the related complex I enzyme and an exploration of its possible proton-translocation network. Within this analysis, a novel, possible proton-entry site was detected. The boundaries and interface regions of the complexes in the supercomplex were analysed to highlight lipid–protein and protein–protein interactions involved in stabilisation/assembly. Overall, the work is of good quality and the improved resolution/definition of the model presents a step forward in NDH-PSI-LHCI supercomplex understanding. I have no major critiques of the work presented, and my comments are focused on comprehension for a more general audience, clarification and expansion on the methodology, and other minor comments.

Title:

The title reads “CryoEM structure of the NDH-PSI-LHCI-2 supercomplex from *Spinacia oleracea*”; however, I find the “NDH-PSI-LHCI-2” nomenclature confusing in this context. I believe the authors are referring to Fig. S3 and are assigning the supercomplex based on the position of PSI relative to NDH. This does not appear to be a consensus in nomenclature across the field and is an internal reference to this paper. This caused confusion as the name is suggestive of NDH-PSI-LHCI with two PSI. Alternatively, LHCI can associated with PSI in certain conditions and I also thought the structure may be of the NDH-PSI-LHCI-LHCI supercomplex from the title alone. My suggestion would be to change to “NDH-PSI-LHCI”, as stated in the introduction.

Introduction:

1. Related to the above, please add a qualifying sentence where the abbreviation “NDH-PSI-LHCI-2” is used to make it clear it is a positional notation outlined in Fig. S3 and not compositional. Furthermore, Fig. S3 is referenced before Fig. S2. Please reorder to be consistent with the text. (pg. 3, line 71)

2. The text reads “In plants and cyanobacteria, NDH takes the role of the mitochondrial NADH:ubiquinone oxidoreductase (complex I). NDH transfers electrons from PSI to the plastoquinone (PQ) pool and cytochrome b6f complex, pumping protons across the thylakoid membrane”, which suggests that plants do not have mitochondrial complex I. Please rephrase to make it clear that the comparison is for an analogy to electron/substrate transfer in the chloroplast. (pg. 2, line 50)

Results

3. “Our cryoEM map of the supercomplex has an average resolution of 3.2 Å”—This is a bit misleading. Please specify in the text this is from four focus refinement maps and not from the consensus averaged global value or provide a range for the focus maps and also the consensus value of 3.3 Å (pg. 3, line 83).

4. “A detergent molecule replacing a lipid facilitates the connection between TMH3 of Lhca-6, the N-terminal stromal loop of PnsB5, and the alpha-helix motif linking TMH1 and TMH2 of NdhF (figure 3H and supplementary figure 7C).”—how do the authors know a lipid is being replaced? Please include a reference for the statement. (pg.4 line 129)

5. In the “The NdhU subunit” section, please describe the role of SubE module and information about the possible roles of the missing subunits in this structure. Are they associated with ferredoxin binding?

6. “In NDH we found densities for 14 phosphatidyl glycerol (PG), 5 monogalactosyl diacylglycerol (MGDG) and 3

sulfoquinovosyl diacylglycerol (SQDG) (supplementary figure 7A), but, surprisingly, no digalactosyl diacylglycerol (DGDG).”—please can the authors expand on why this is surprising. Do the known lipid compositions of the membrane suggest DGDG is in high abundance? (pg.5 line 162). Do the authors have a comment on the importance of cardiolipin in the mitochondrial membrane for complex I activity, yet its absence here in the thylakoid membrane with the related NDH enzyme? Do they suspect a redundancy in lipid composition, a lack of importance for NDH, or some other reason?

7. “The PQ sits at the entrance of the cavity surrounded by several charged residues, interacting with Y259 of NdhA through its ketyl group (figure 7C).” (pg.6 line 196). I believe “ketyl” is incorrect here. It should be carbonyl or ketone.

8. “Although we cannot rule out t-PCC α M (4-trans-(4-trans-propylcyclohexyl)-cyclohexyl α -maltoside, the detergent used for solubilising and purifying the supercomplex), the density in the PQ pocket looks too long and bulky for a detergent molecule, and PQ fits it almost perfectly (figure 7C).” (pg.6 line 199) Outside of visual speculation, was the detergent docked into the region and compared to the PQ for best fit? Map–model correlation values?

9. Serine is misspelled “serin” (pg.8 line 240).

10. “A t-PCC α M detergent and one MGDG (MGDG 1016/F, figure 8D) contribute to the formation of the funnel by blocking the lateral exit, interacting with NdhB, NdhE, the transversal helix of NdhF and PnsB1 (figure 6D, supplementary figure 7A, C).” (pg.8 line 257)—given the inclusion at this site of the detergent used in the purification, how trustworthy is funnel region definition in relation to the non-solubilised complex?

Discussion:

11. “Our findings support previous studies indicating that particles with a 1:1 NDH to PSI-LHCI ratio predominantly preserve PSI-LHCI-2” (pg.10 line 307). In relation to my earlier point with this nomenclature, please specify here that “PSI-LHCI-2” is in reference to the position of PSI relative to NDH.

12. Given the point of contact of the PSI to the NDH is predominantly from Lhca6 to NdhF, do the authors have a rationalisation for why this limited interaction of the PSI at this site is stabilising for the whole of the NDH, where pretty much all of subunit interactions are independent of PSI? For example, the “PSI-LHCI-1” position (Fig. S3) has far more contact with NDH. The authors highlight the predominance of this PSI position in the 1:1 supercomplex in the literature. Can they suggest other possible reasons (outside of NDH stability) for why this interaction would occur? Has a role for 1:1 proximity to ensure efficient cycling through membrane complex distribution been suggested in the literature?

Model:

13. I noticed in the model that the coordination geometry of the FB and N6b clusters to some of their respective Cys ligands (Cys17 and Cys110) is a little off. As these are not the focus of the paper, I think it is okay, but be cautious in the future and consider imposing stronger restraints in regions of lower resolution that have FeS clusters.

Figures:

14. Figure 8. “(D) Close-up of the constriction in channel NdhB-D.”—please check. Should this be (G)? Add reference in the legend at G to the asterisk in this panel also.

15. Figure 8 A, B, E and G all show grey surface as the detected channel; however, D and F have the sliced view of the detected cavities (is this correct? —please add to legend if so). Please reconsider the colour choice/inclusion for the latter for D and F as it tends to obscure the information rather than help. Consider the simple model view or VDW radii or model surface representation in trying to visualise the cavity in the protein as the cavity surfaces are already represented in A and E.

16. Figures S10, 12 and 14 are referenced in the text before S9 (and S11 and S13). Reorder figures to reflect appearance in the text.

17. Supplementary figure 6 is referenced for the first time in the Discussion. This should be renumbered to follow the text citations (after current Fig S10, S12 and S14).

18. For supplementary figure 6, please state which cryoEM map is displayed in F (this work or EMBD 32477?).

19. Supplementary figures 17 and 18 image labels read “wast” instead of “waste”.

20. Supplementary figure 7 and 8 (and others): please add to the legend number and letters above the lipids/detergents indicated refer to the model chain IDs (I am assuming). This identifier nomenclature seems to be used throughout the manuscript so it may be worth adding a sentence to this effect in the main text somewhere.

21. Supplementary figure 9: please add the threshold value for the map for the displayed map features and state which map was used (e.g., composite). Were there no map features in the PQ cavity (outside of the proposed PQ density)?

22. Supplementary figure 10 legend sentence repeated: “. An * (asterisk) indicates positions which have a single, fully conserved residue. An asterisk indicates positions of single, fully conserved residues.”

23. Supplementary figures 11–16 do not have the figure organism name “*S. oleracea*” in italics.

Tables:

24. Supplementary table 2 is not referenced in the main text. Please add to a suitable location or consider removing it if not needed.

25. Please reference supplementary table 3 in the methods near PHENIX/Molprobit.

Methods:

26. “...in electron counting mode at a calibrated pixel size of 0.837 Å” (pg.12 line 386)—was this a facility calibration or one done with your own references? If the latter please provide additional details i.e., what reference model.

27. “55.704 dose fractionated movies” (pg.12 line 386)—incorrect full stop placement: “55,704”

28. “Details are shown in supplementary figures 17 and 18.” (pg.12 line 389). This references the purification supplementary figures in the cryoEM data collection section. I believe this may be incorrect placement. S19 instead, perhaps?

29. Please add a reference for cryoSPARC (10.1038/nmeth.4169) outside of the specific reference for 3DVA already included.

30. The Relion reference only refers to the Bayesian polishing improvement. Consider adding the reference to Relion-3 (as suggested was used in the reporting summary), which contains advancements such as CTF refinements (that were also carried out by the authors) (10.7554/eLife.42166).

31. Please provide information about where the real-space mask came from, Auto-generated? For the focus maps, please provide details for how these were made (molmap/volumes, extensions/dilations, soft edges). Consider adding the mask outlines to supplementary figure 2.

32. Please verify you mean to cite UCSF Chimera and not UCSF ChimeraX (the reporting summary states ChimeraX v1.2). The latter is used for ISOLDE, so it should be included as a citation anyway. Please use the citation most in line with the version of ChimeraX used (I think it is probably 10.1002/pro.3235). Remove UCSF Chimera reference if this was not used (not listed in the reporting summary).

33. The reporting summary suggested the version of PHENIX used was much later than the reference used for the citation; consider updating the reference to something more current (doi:10.1107/S2059798319011471).

34. Please provide the probe size used in the HOLLOW software to generate the tunnels. This will help with comparisons to other software.

35. You started with an initial blob pick of ~17 million particles, of these if got narrowed down to ~4.5 million particles. Subsequently, these were refined to 45k and 168k particles, before being grouped to 107,752, cleaned again to 88,276 before a final classification to 38,385 particles. The final particle number is an incredibly small percentage of the starting material. The authors should provide more information on the types of classes that are dismissed. Are they all junk particles? Are they of varying compositions? Fig. S19 should be expanded to include information about examples of dismissed volumes at the E and F stages, which would be more informative than repeating the same final volume from each Topaz picking pipeline. There is space on the figure for both. In the CN-PAGE gel you demonstrate presence of isolated PSI and CF1Fo. Were these detected in the cryoEM analysis also?

36. “3D-refined particles with a real-space mask around the peripheral arm were used as inputs for 3D variability analysis (3DVA) in order to separate particles that showed a well-defined peripheral arm from those where it was damaged.” (pg.13 line 403). Please better define “damaged”. Low-resolution regions (local resolution maps)? Missing subunits?

37. “Subsequently, the models were docked into the various maps using UCSF Chimera” (pg.13 line 421). Please specify what the “various maps” were. Each respective focus map used to build that specific region?

38. “Initial model fitting was conducted using ISOLDE67, and iterative refinement was performed with Coot66 in conjunction with PHENIX”. (pg.13 line 423) Please specify which map was used for the final refinement i.e., the consensus cryoSPARC map or the composite map.

39. Please add the Rama-Z scores from PHENIX validation to Supplementary Table 3.

References:

Please be more careful about referencing the appropriate resource for cryoEM/model building software that you used (examples noted above).

Authors should review the references in the bibliography and take more care with the reviewing the formatting. Examples of

issues are non-italicised organisms and unwanted characters such as it and * in titles (refs 67, 68, 13, 17, 18, etc.)

General:

Is there any functional data (here or in the literature) to support that the sample in the structural analysis can perform its predicted function (with respect to the absent subunits)?

Are there any mutants with knock outs (in related model organisms) of the absent subunits in this structure and how do they behave? This may inform whether the missing subunits are necessary for catalysis. NdhT and NdhU look very structurally similar. Do authors predict redundancy in these subunits at the modelled position? A possible role as an assembly factor? A brief comment in the Discussion on the predicted differences between this structure and the native structure with these included 'missing' subunits may be useful.

Why the choice of this detergent over others?

Version 1:

Decision Letter:

Our ref: NSMB-A49056A

6th Aug 2024

Dear Dr. Kühlbrandt,

Thank you for submitting your revised manuscript "CryoEM structure of the NDH-PSI-LHCI supercomplex from *Spinacia oleracea*" (NSMB-A49056A). It has now been seen by the original referees and their comments are below. The reviewers find that the paper has improved in revision, and therefore we'll be happy in principle to publish it in *Nature Structural & Molecular Biology*, pending minor revisions to satisfy the referees' final requests and to comply with our editorial and formatting guidelines.

Thank you again for your interest in *Nature Structural & Molecular Biology*. Please do not hesitate to contact me if you have any questions.

Sincerely,
Kat

Katarzyna Ciazynska, PhD
(she/her)
Associate Editor
Nature Structural & Molecular Biology
<https://orcid.org/0000-0002-9899-2428>

Reviewer #1 (Remarks to the Author):

Second round review of "CryoEM structure of NDH-PSI-LHCI supercomplex from *Spinacia oleracea*".

The authors took the effort to address all issues and questions raised by us. Their corrections and explanations, both in the Manuscript as well as in their "Response to reviewers" are fully satisfactory. Thank you also for the non-essential clarifications.

We recommend the work for publication.

TYPOS:

- In the main text "Supplementary figure(s)" is spelled with both capital and small "S".

Reviewed by Ricardo Righetto and Wojciech Wietrzynski

Reviewer #2 (Remarks to the Author):

The authors have adequately addressed all of the reviewers comments and no further revisions are needed.

Reviewer #3 (Remarks to the Author):

The authors have addressed all aspects that were brought up in the review process well.

Minor comments: Supplementary figure 1 has a repeated 'from' in its title and Supplementary Table 2 is lacking a space in the 'Datacollection and processing' column header.

Version 2:

Decision Letter:

17th Dec 2024

Dear Dr. Kühlbrandt,

We are now happy to accept your revised paper "CryoEM structure of the NDH-PSI-LHCI supercomplex from *Spinacia oleracea*" for publication as an Article in Nature Structural & Molecular Biology.

Your paper will be published online soon after we receive proof corrections and will appear in print in the next available issue. You can find out your date of online publication by contacting the production team shortly after sending your proof corrections.

If you have not already done so, we strongly recommend that you upload the step-by-step protocols used in this manuscript to the Protocol Exchange. Protocol Exchange is an open online resource that allows researchers to share their detailed experimental know-how. All uploaded protocols are made freely available, assigned DOIs for ease of citation and fully

searchable through nature.com. Protocols can be linked to any publications in which they are used and will be linked to from your article. You can also establish a dedicated page to collect all your lab Protocols. By uploading your Protocols to Protocol Exchange, you are enabling researchers to more readily reproduce or adapt the methodology you use, as well as increasing the visibility of your protocols and papers. Upload your Protocols at www.nature.com/protocolexchange/. Further information can be found at www.nature.com/protocolexchange/about.

Please note that *Nature Structural & Molecular Biology* is a Transformative Journal (TJ). Authors may publish their research with us through the traditional subscription access route or make their paper immediately open access through payment of an article-processing charge (APC). Authors will not be required to make a final decision about access to their article until it has been accepted. [Find out more about Transformative Journals](https://www.springernature.com/gp/open-research/transformative-journals)

Sincerely,

Katarzyna Ciazynska, PhD
(she/her)
Senior Editor
Nature Structural & Molecular Biology
<https://orcid.org/0000-0002-9899-2428>

We thank the reviewers for their thorough revision of our manuscript, their insightful comments and valuable suggestions, which we have incorporated. Below please find a point-by-point response to the reviewers' comments. Changes in the revised manuscript "2024-07-03-NSMB_INTROINI_REVISIED_MANUSCRIPT" are highlighted in yellow.

Reviewer #1:

General comments:

The authors present the cryo-EM structure of the NDH-PSI-LHCI-2 supercomplex from spinach at a global resolution of 3.2 Å. It is a third recent work describing a similar NDH-PSI-LHC supercomplex from vascular plants. Here, authors manage to obtain a higher overall resolution of the supercomplex than previously reported, providing insights into subunit interactions and cofactors unresolved in other organisms. The work is a purely structural study. Results are exhaustive and clearly presented overall. Points that need further clarification are outlined below. The comparative approach, both with homologous models of NDH-PSI-LHCI complexes and mitochondrial Complex I is much appreciated.

- Novel biological insight focuses on a more precise description of the NDH complex, including assignment of the NdhU subunit and the first elucidation of the full PnsB1 subunit and its interactions within the supercomplex. However, the discussion of the potential missing iron-sulfur cluster in the Pnsb3 subunit is lackluster. Can it be the effect of the different purification protocol or have a physiological meaning? Could authors comment on that?

Thank you for bringing up this point. As discussed on page 10 of the manuscript, we attribute the absence of the iron-sulfur cluster in this position to differences in purification protocols between our study and previous work by Su et al. (2022) and Shen et al. (2021).

As stated on page 10 of our manuscript: “Even though we isolated the supercomplex under reducing conditions, some cysteines in subunit PnsB3 formed disulphide bonds (Extended Data Fig. 5a-d), in contrast to the *Arabidopsis*⁸ and barley⁹ supercomplexes that were purified without reducing agents but show an Fe-S density in this position.”

We prefer not to speculate on the physiological significance of this potential iron-sulfur cluster, as further investigations would be needed to establish its presence and function in the complex.

- Second, from the overall calculation, chlorophyll and carotenoids content per PSI-LHCI of the NDH-PSI-LHCI-2 reported here seems greater (154 and 38 versus (includes two PSI-LHCI) 298 and 67 respectively) than that in the NDH-PSI-LHCI-1 supercomplex from barley. Previous studies reported different Chl content of Lhca5

and Lhca6 subunits compared to other LHCI and counts varied between species. This should be clarified in the text.

The difference in the number of chlorophylls in barley (Shen et al., 2021) and our model (see legends in figure 1) is due to three factors: first and foremost, the resolution of our map is substantially higher (EMDB-19248: 3.09Å) compared to barley PSI-LHCI-2 (EMDB-31350: 3.88Å) and *Arabidopsis* PSI-LHCI-1 (EMDB-31348: 3.40Å). At a lower resolution of 3.4-3.9 Å, pigment densities are easily missed or mistaken. Second, as suggested by the reviewer, the two copies of PSI-LHCI 1 and 2 interact with NDH via two different subunits (Lhca-5 and Lhca-6) and this may affect the position and number of bound chlorophylls. Third, PSI-LHCI from barley is missing the subunit PsaG that contains several chlorophylls, as is the case in the *Arabidopsis* complex.

The comparison of our PSI-LHCI-2 structure to PSI-LHCI-1 from barley shows that our map contains six additional chlorophylls and four more carotenoids (see black squares a-f in Figure 1d below). Comparison with the better-resolved model of PSI-LHCI-1 from *A. thaliana* (Su et al., 2022, EMDB- 32463, 3.3 Å) indicates a difference of only two chlorophylls. As shown in Figure 1d and e below, the chlorophylls labelled a and b in the figure are consistently absent in PSI-LHCI-1 from barley and *A. thaliana* when compared to our PSI-LHCI-2. Another clear difference between our PSI-LHCI-2 structure and both published PSI-LHCI-1 structures is an extra chlorophyll between Lhca-2 and Lhca-5 (see red square z in Figure 1d and e). The missing or extra chlorophylls in PSI-LHCI-1 are close to Lhca-5 (the equivalent of Lhca-6) that is involved in the interaction with NDH. Thus, we assume that their presence or absence may be related to energy transfer in PSI-LHCI-1.

The apparent absence (Figures 1a and d, chlorophylls labelled c-d) of chlorophylls in barley compared to our model might be due to the lower resolution of the barley map in these peripheral positions (Figure 1a and 1d letter d and e) and because of the absence of PsaG subunit (Figure 1a and d, letter c). On the other hand, some chlorophylls are visualized in barley in different positions compared to spinach (Figure 1a letters x, y, 1d letters x, y). However, all together these disparities in the number and position of chlorophylls in barley may be species-specific. These differences may thus reflect the long evolutionary distance between barley, a monocotyledon, and spinach and *A. thaliana* which are dicotyledons.

Figure 1: New Supplementary Figure 1: Comparison of PSI-LHCI supercomplexes from spinach, barley and *A. thaliana*. We refer to the complexes as PSI-LHCI-1 or PSI-LHCI-2 as specified in Extended Data Fig. 2. Chlorophylls in PSI-LHCI-2 from *S. oleracea* compared to PSI-LHCI-2 (a) from barley and (c) *A. thaliana*, and to PSI-LHCI-1 (d) from barley and (e) from *A. thaliana*. (c) The superposition of PSI-LHCI-2 from spinach (cartoon, helices shown as cylinders) with PSI-LHCI-2 from barley (green-yellow surface) reveals that subunit PsaG is missing in barley (PsaG is missing also in PSI-LHCI-1 from barley. (a, d, e) Red squares highlight extra chlorophylls that are found in barley and *A. thaliana* but not in spinach; black squares indicate the position of chlorophylls observed only in spinach. Another persistent difference between our PSI-LHCI-2 and both PSI-LHCI-1 is an extra chlorophyll between Lhca-2 and Lhca-5 (red square z in supplementary figures d and e). The absent or extra chlorophylls in PSI-LHCI-1 are close to Lhca-5 (the equivalent of Lhca-6) which is involved in the interaction with NDH. Thus, we assume that their presence or absence in this position may be related to energy transfer in PSI-LHCI-1.

We have modified the text **on page 4** of the revised manuscript as follows:

“In our PSI-LHCI-2 density map, we observed 154 chlorophylls, matching the count in PSI-LHCI-2 from *A. thaliana*, but exceeding the 148 chlorophylls in PSI-LHCI-2 from *H. vulgare* (supplementary figure 1a, c). Additionally, PSI-LHCI-2 from *S. oleracea* has more

chlorophylls than PSI-LHCI-1 from both *A. thaliana* (152) and barley (148) (supplementary figure 1d, e). This discrepancy is most apparent in subunit Lhca-5, which in PSI-LHCI-1 interacts with NDH (see chlorophylls labelled a, b and z in supplementary figure 1d, e), suggesting that differences in pigment content reflect the variation in subunit composition and that the presence or absence of particular chlorophylls might be required for energy transfer in PSI-LHCI-1.”

We have added a new **supplementary figure 1** (see above) and revised the **discussion** as follows:

“Our PSI-LHCI-2 contains the same number of chlorophylls (154) as PSI-LHCI-2 from *A. thaliana* but more than PSI-LHCI-2 from *H. vulgare* (148) (supplementary figure 1a, c). We attribute this difference first and foremost to the significantly higher resolution of our map (EMDB-19248: 3.09Å) compared to barley PSI-LHCI-2 (EMDB-31350: 3.88Å). At a resolution of around 3.9Å, pigment densities are not very clear and can be easily missed or mistaken. Second, both PSI-LHCI-1 or 2 complexes from barley lack the subunit PsaG (supplementary figure 1c), which, as in our spinach structure and in *A. thaliana*, contains several chlorophylls and carotenoids. Certain chlorophylls are observed only in barley (supplementary figure 1a, d highlighted with letter x and y). These disparities in the number and position of chlorophylls in barley may be species-specific. This may reflect the evolutionary distance between barley, a monocotyledon, and spinach or *A. thaliana*, which are dicotyledons.”

- While the cryo-EM imaging and data processing methods are solid (pending only minor clarifications, see below), the modeling of the atomic structure needs some reevaluation. As evidenced by the structural validation report, the overall model quality does not seem to be on par with the map quality. In particular, the clash score and the sidechain outliers stick out. One cycle (or a few) of real-space refinement in PHENIX should be able to fix most issues automatically at this stage. Care should be taken in particular with the nonbonded_weight parameter (the default value of 100 in PHENIX is too low for typical cryo-EM maps, especially of large complexes) and the rotamer fitting procedures (e.g. local grid search should help). The authors should nevertheless carefully reexamine the model and its fit to the experimental density afterwards, to evaluate any significant changes that could affect their conclusions, and remake figures when necessary.

Thank you for this suggestion. We re-refined the model and the composite map with the latest release of PHENIX, which resulted in a new model with improved scores. Figure 2 is a screenshot of the PHENIX summary. After careful re-examination we made the necessary modifications to the figures if there were significant changes. Overall, however, our conclusions remain unaffected.

The file named “D_9100095518_val-report-full_P1.pdf” is the validation report for the new PDB file. This report was downloaded from the PDB validation server due to issues we are

encountering with the file upload system on the wwPDB deposition system. We are currently in contact with wwPDB support, and they are working to resolve the issue.

Figure 2: Screenshot from PHENIX summary of the new model validated against the composite map.

Specific comments:

- L390 Image processing for single particle cryo-EM
The authors should explain their rationale for importing their project from cryoSPARC into RELION for specific refinement tasks (3D refine, CTF refine, Bayesian polishing) and back into cryoSPARC.

We imported particle coordinates from cryoSPARC to RELION to perform CTF refinement and Bayesian polishing. Subsequently, we conducted 3D refinement of the particles in both RELION and cryoSPARC, ultimately achieving a better final result in cryoSPARC.

We re-wrote this passage which now reads: “Particle coordinates were exported to Relion⁶³ and underwent 3D refinement, multiple rounds of CTF refinement and Bayesian polishing. Subsequent 3D refinement jobs were conducted in both RELION and cryoSPARC, with cryoSPARC yielding superior final results. Next, these particles were subjected to heterogeneous refinement, isolating a subset of 88,276 particles.”

- L416 Model building and analysis

Figure 1:

The current choices of color and lighting make it difficult to understand the architecture and details of this intricate cryo-EM map. We suggest the authors experiment with ambient occlusion lighting (for example as implemented in ChimeraX) and surface outlines to improve interpretability. Furthermore, the labels indicating the different proteins and domains are difficult to read, a font outline could be helpful here as well.

We improved the color scheme and modified the font in Figure 1.

- Figure 4G:
The densities for Y222 and F232 are not well resolved in comparison to other side chains in the same helix, at least at the isosurface contour shown; could the authors comment on it? For example, are there nearby interactions (or lack thereof) affecting the resolution of these densities?
(See also general comment about model quality above).

Thank you for this suggestion. The sidechains did not show up clearly at the high contour level used to create the image (0.425). We have now re-drawn Figure 4 at a lower contour level of 0.3. In addition, we have specified the contour level used for visualizing the map in each panel.

- Supp. Fig 2:
The FSC plots should be bigger; they are difficult to evaluate as they are currently. While we acknowledge there is no defined standard in the field for how local-resolution maps should be presented, we suggest that reversing the color-coding would be more intuitive: red for lower resolution parts (more disordered, “hot”) and blue for the highest-resolution parts (well-ordered, “cold”) of the map. This is not essential and should only be done at the authors’ discretion.
The model-map FSC curve should also be presented, in this figure or elsewhere.

We have updated Extended Data Fig. 4 with the suggested color-coding for the local resolution maps and increased the size of the FSC plots.

In addition, we created a new figure (Extended Data Fig. 3) that includes the globally refined map with its corresponding FSC curve and a diagram showing the angular distribution of particles. This figure also presents the composite map and the model used for validation, accompanied by the validation slider and the map-versus-model FSC curves, with and without masking, calculated in PHENIX.

- Supp. Fig. 19:
This figure should be “closer” to Supp. Fig. 2 (either before or after it) as they are directly related.
The figure panels should be rearranged so that the example micrographs and 2D

classes can be shown bigger; it is quite difficult to evaluate them currently. A common issue in single-particle cryo-EM, in particular for membrane protein complexes, is preferential orientation. The authors should include an angular distribution plot to aid interpretation of their experimental density.

We have reorganized the order of our Extended Data Figures and Supplementary Figures to match their citation in the text. We have divided “Supp. Fig. 19” into two separate figures, now labeled “Supplementary Figure 9” and “Supplementary Figure 10”. This allowed us to enlarge both the micrographs and 2D classes (see Supplementary Figure 9). As mentioned above, we have incorporated the angular distribution plot into the new Extended Data Fig. 3.

- Supp. Table 2:
The model-map resolution from the FSC should be evaluated at the FSC=0.5 threshold, as it is comparing a noiseless structural model to a noisy map (see Rosenthal & Henderson 2003 and Phenix validation documentation for details).

Thank you. We have modified the table as suggested.

Minor comments:

- The Discussion section is basically a repetition of the results and conclusions drawn by authors and others; especially paragraphs comprising L297-L318 and L330-L339. E.g.: L297-L298: “Our structure shows that Lhca-6 interacts with the SubB module of NDH, confirming earlier speculations” – That interaction was quite firmly confirmed before.

We are aware that previous studies have established this interaction, primarily through biochemical assays and cryoEM density maps with limited resolution, preventing a detailed assessment of the supercomplex organization. We have revised the discussion sentence at line 319 accordingly: “Lhca-6 of PSI is known to be required for interaction with NDH^{6,50}. Our study offers clear evidence that Lhca-6 interacts with the SubB module of NDH, whereas earlier conclusions^{28,51} were speculative.”

- In our opinion it would be more interesting for the reader to speculate on differences between NDH-PSI-LHCI-1 and -2 supercomplexes reported in the literature. The authors are rare experts in the physiological-structural studies; their comments on points such as the importance of two-point stabilization of the NDH complex by the PSI-LHC1s or potential heterogeneity of the supercomplexes in vivo could be particularly valuable.

Thank you. We have made added some comments to the discussion at line 321 but we prefer not to speculate as further investigations would be needed.

Discussion, page 10, line 319:

“Lhca-6 of PSI is known to be required for interaction with NDH^{6,50}. Our study offers clear evidence that Lhca-6 interacts with the SubB module of NDH, whereas earlier conclusions^{28,51} were speculative. Some studies⁷⁻⁹ found that NDH can bind a second PSI-LHCI assembly (i.e. PSI-LHCI-1), through interactions of the PnsB1 β 3- β 5 and β 5- β 6 loops and the Lhca-5 TMH2-3 loop (Extended Data Fig. 2b, c and Extended Data Fig. 10b-e). Sequence alignments of spinach PnsB1 and Lhca-5 with the corresponding subunits in *A. thaliana* and *H. vulgare* showed no significant amino acid differences in the interacting interface, ruling out sequence variations as the cause for the observed 1:1 ratio of NDH to PSI-LHCI (Extended Data Fig. 10g-i). The abundance of interactions between Lhca-6 and NDH (Fig. 3c-h) potentially stabilizes the formation of the smaller supercomplex, while transient interactions may favor formation of the larger supercomplex with two PSI assemblies^{8,9}. Our findings support previous studies indicating that particles with a 1:1 NDH to PSI-LHCI ratio predominantly preserve PSI-LHCI-2⁷⁻⁹. PSI-LHCI-2 refers to the position of PSI relative to NDH as illustrated in Extended Data Fig. 2. The higher stability of the smaller supercomplex may be the result of evolutionary adaptation, suggesting that angiosperms evolved to stabilize the NDH complex by forming the Lhca-6-dependent supercomplex instead of producing new NDH complexes in mature leaves²⁸.”

Immunoblotting and mass spectrometry consistently detected Lhca-6 in the NDH-PSI-LHCI-2 supercomplex⁶, but not in PSI monomers¹¹, suggesting a key role of Lhca-6 in supercomplex formation. However, variations in the number of bound PSI-LHCIs may be depend on environmental factors or leaf maturity^{52,53}. Further investigation is needed to determine if the reversible formation of the supercomplex is dependent on environmental stress, light intensity, or leaf maturity”.

Typos and others:

- We have been recently informed by one of the Nature Publishing Group editors to avoid the term “higher plant”. We propagate this information to the authors.

Thank you. We have replaced the term “higher plants” by “angiosperms”.

- “Lhca-6” appears in the text also as “Lhca6” - is that intentional?

This was a mistake, thank you for pointing it out.

- L183: crystal structure of in *P. sativum* PSI-LHCI
We have removed “in”.

- L377: screed (screened)

Corrected, thanks.

- L728: Figure 4: Identification NdhU (Identification of NdhU)

Corrected.

Reviewed by:

Ricardo D. Righetto and Wojciech Wietrzynski

Reviewer #2:

Remarks to the Author:

In their manuscript “CryoEM structure of the NDH-PSI-LHCI-2 supercomplex from *Spinacia oleracea*” Introini et al. present the structure of the titular photosynthetic supercomplex at significantly improved resolution compared with previous work. This allows the author to build models at near atomic resolution for the entire supercomplex and identify additional subunits that were lacking in previous reconstructions and models. Additionally, the authors are able to use their improved models to compare and contrast the structural features of the chloroplast NDH complex to structures from cyanobacteria and respiratory complex I. The approach and data are valid and of high quality. Altogether the paper is very well written and will be of wide interest to those in the field of bioenergetics, plant physiology and membrane protein structural biology.

Strengths:

- 1) improved structure of the NDH-PSI-LHCI-2 supercomplex allow for the modeling of previously low-resolution domains.
- 2) detailed characterization of interactions between NDH and PSI-LHCI-2 complexes.
- 3) identification of the J-like NdhU subunit.
- 4) identification of several lipid binding sites that are conserved in homologous complexes.
- 5) description of the aqueous half channels and hydrophilic axis in the membrane domain.
- 6) comparison of NDH to respiratory complex I.

- The paper could be improved by a discussion in the intro about the relative abundance of the NDH complex. Relative to linear photosynthesis how much flux is known to occur through the cyclic electron flow (CEF) pathway and is the abundance of the NDH complexes in thylakoids sufficient to accommodate the levels of flux observed. There is controversy in the literature regarding the degree to which NDH contributes to CEF (see pmid: 30827891 for example) as it is relatively low abundance (1:100 ratio of NDH:PSI in tobacco leaves; pmid: 9463365) and low flux ($2.5 \text{ e}^-/\text{s} \cdot \text{PSI}$ compared to CEF flux of $60 \text{ e}^-/\text{s} \cdot \text{PSI}$ measured in *Chlamydomonas*; pmid: 30498023, 30827891) . How can the structure presented here inform on this controversy.

In our opinion, our current work does not allow us to speculate on this controversy. To draw meaningful conclusions, additional experiments would be necessary, e.g. biochemical assays to confirm the functional roles of specific subunits, comparative studies across different species, or cryoEM structures at even higher resolution, which will however be difficult and is outside the scope of the present work.

Minor points:

- Page 2, line 50: the authors write “NDH takes the role of the mitochondrial NADH:ubiquinone oxidoreductase (complex I)”, however, given that the reaction it catalyzes is different this language is too strong, perhaps change to “NDH is analogous to...”

Changed to: "In thylakoid membranes of plant chloroplasts and cyanobacteria, NDH is homologous to the mitochondrial NADH:ubiquinone oxidoreductase (complex I)".

- Page 2, line 58: the authors refer to the ubiquinone reduction (Q) domain of complex I” it is more common to refer to this region of complex I as the “Q module” as opposed to domain and the authors do so later in the text, good to be consistent throughout.

Line 52, modified as follows: “Subunits NdhH-K in the peripheral arm of NDH are homologous to the ubiquinone reduction (Q) module of complex I (NDFUS2,3,7,8, Extended Data Fig. 1)”

- Page 3, line 87: the authors state that “We detected 26 out of the 29 NDH subunits...” but do not indicate how these were “detected” are they referring to what they are able to model in the reconstruction or detect on a gel or by mass spec?

The subunit identities were confirmed by our cryoEM density map. The text at line 83 has been revised as follows: “Our cryoEM density map revealed 26 out of the 29 NDH subunits¹⁰ and all 16 subunits of the PSI-LHCI-2 complex...”

- Page 4, line 130: the author state “A detergent molecule replacing a lipid...” but do not state how they determined that this site would normally be occupied by a lipid in thylakoids, given that no lipid is observed in their structure at this position perhaps this phrase should be made more speculative, i.e., “likely replacing a lipid...”

Text revised as follows: “A detergent molecule, likely replacing a lipid, appears to reinforce the connection between TMH3 of Lhca-6, the N-terminal stromal loop of PnsB5, and the alpha-helix motif linking TMH1 and TMH2 of NdhF (Fig. 3h and Extended Data Fig. 7a).”

- Supplementary figure 6 is not referred to in the text.

Thanks, included.

- Page 6, line 191: the authors refer to “the PQ entry channel, E channel and PQ binding pocket of complex I” however, complex I utilizes ubiquinone not plastoquinone.

Text revised as follows at line 195: “At the juncture of the membrane arm and the peripheral arm the HOLLOW software tool³⁵ revealed a bifurcated cavity. This cavity comprises the PQ entry channel, E channel, and PQ binding pocket, closely resembling the homologous ubiquinone (Q) entry channel, E channel, and Q binding pocket of complex I (Fig. 1d and 7a, b).”

- Supplementary figure 1: “Those that are conserved are...” is a bit ambiguous, perhaps rewrite “Conserved modules are...”

Revised as follows: “Extended Data Fig. 1: NDH is the chloroplast homolog of complex I. Lateral views of (a) and (c) *S. oleracea* NDH and (b) and (d) *A. thaliana* complex I (PDB ID 8BEF, 8BEH, 8BED). The different modules forming those complexes are colour coded. Conserved modules are shown in the same colour in both models.”

- Supplementary figure 2: The FSC plots are too small and need to be made larger with larger font.

We have revised Extended Data Fig. 4 and increased the size of the FSC plots.

- Supplementary figure 9, line 959: the authors refer to “the continuous aqueous passage...” however according to the figure the aqueous passage is not continuous, perhaps “continuous hydrophilic passage...” is what was meant?

We have revised Extended Data fig. 8 didascaly as follows: “The continuous aqueous passage, defined by hydrophilic residues from the Q-binding site to NdhF, is indicated by the light-blue arrow.”

Reviewer #3:

Remarks to the Author:

The authors present the *Spinacia oleracea* NDH-PSI-LHCI supercomplex at an improved resolution relative to previous publications on this supercomplex (and the related NDH-PSI2-LHCI) of different species, in part due to using focus refinements on sub-regions. The authors used focused classification to better resolve the peripheral domain of the NDH complex, resulting in improved definition of elements that compose it and the identification of the NdhU subunit at this site also. The improved resolution of NDH allowed for the modelling of

the N-terminal region of PnsB1, and a comparison to the related complex I enzyme and an exploration of its possible proton-translocation network. Within this analysis, a novel, possible proton-entry site was detected. The boundaries and interface regions of the complexes in the supercomplex were analysed to highlight lipid–protein and protein–protein interactions involved in stabilisation/assembly. Overall, the work is of good quality and the improved resolution/definition of the model presents a step forward in NDH-PSI-LHCI supercomplex understanding. I have no major critiques of the work presented, and my comments are focused on comprehension for a more general audience, clarification and expansion on the methodology, and other minor comments.

Title:

The title reads “CryoEM structure of the NDH-PSI-LHCI-2 supercomplex from *Spinacia oleracea*”; however, I find the “NDH-PSI-LHCI-2” nomenclature confusing in this context. I believe the authors are referring to Fig. S3 and are assigning the supercomplex based on the position of PSI relative to NDH. This does not appear to be a consensus in nomenclature across the field and is an internal reference to this paper. This caused confusion as the name is suggestive of NDH-PSI-LHCI with two PSI. Alternatively, LHCII can associated with PSI in certain conditions and I also thought the structure may be of the NDH-PSI-LHCI-LHCII supercomplex from the title alone. My suggestion would be to change to “NDH-PSI-LHCI”, as stated in the introduction.

Title changed as suggested.

Introduction:

1. Related to the above, please add a qualifying sentence where the abbreviation “NDH-PSI-LHCI-2” is used to make it clear it is a positional notation outlined in Fig. S3 and not compositional. Furthermore, Fig. S3 is referenced before Fig. S2. Please reorder to be consistent with the text. (pg. 3, line 71)

Text revised as follows: “Here, we present the structure of the complete NDH-PSI-LHCI supercomplex from *S. oleracea* at 3 to 3.3Å resolution (Extended Data Fig. 3b and 4). We refer to this complex as NDH-PSI-LHCI-2 (see Extended Data Fig. 2).

2. The text reads “In plants and cyanobacteria, NDH takes the role of the mitochondrial NADH:ubiquinone oxidoreductase (complex I). NDH transfers electrons from PSI to the plastoquinone (PQ) pool and cytochrome b6f complex, pumping protons across the thylakoid membrane”, which suggests that plants do not have mitochondrial complex I. Please rephrase to make it clear that the comparison is for an analogy to electron/substrate transfer in the chloroplast. (pg. 2, line 50)

Revised at line 45: "In the thylakoid membranes of plants and cyanobacterial chloroplasts, NDH is homologous to the mitochondrial NADH:ubiquinone oxidoreductase (complex I)."

Results

3. “Our cryoEM map of the supercomplex has an average resolution of 3.2 Å”—This is a bit misleading. Please specify in the text this is from four focus refinement maps and not from the consensus averaged global value or provide a range for the focus maps and also the consensus value of 3.3 Å (pg. 3, line 83).

Text revised as follows: “Our cryoEM map of the supercomplex has an average resolution of 3.2 Å (according to local refined maps resolutions, Extended Data Fig. 4), but well-ordered regions are locally resolved up to 3Å (Extended Data Fig. 4).”

4. “A detergent molecule replacing a lipid facilitates the connection between TMH3 of Lhca-6, the N-terminal stromal loop of PnsB5, and the alpha-helix motif linking TMH1 and TMH2 of NdhF (figure 3H and supplementary figure 7C).”—how do the authors know a lipid is being replaced? Please include a reference for the statement. (pg.4 line 129)

Revised as follows: “A detergent molecule, likely replacing a lipid, appears to reinforce the connection between TMH3 of Lhca-6, the N-terminal stromal loop of PnsB5, and the alpha-helix motif linking TMH1 and TMH2 of NdhF (Fig. 3h and Extended Data Fig.7a).”

5. In the “The NdhU subunit” section, please describe the role of SubE module and information about the possible roles of the missing subunits in this structure. Are they associated with ferredoxin binding?

Revised as follows: “Biochemical assays^{12,29,30} and cryoEM structures of cyanobacterial NDH^{29,30} both revealed that SubE is necessary for the interaction of the complex with Ferredoxin.”

For a more complete answer regarding the role of SubE subunits and possible role of NdhU see the last section of our reviewer response under "General".

6. “In NDH we found densities for 14 phosphatidyl glycerol (PG), 5 monogalactosyl diacylglycerol (MGDG) and 3 sulfoquinovosyl diacylglycerol (SQDG) (supplementary figure 7A), but, surprisingly, no digalactosyl diacylglycerol (DGDG).”—please can the authors expand on why this is surprising. Do the known lipid compositions of the membrane suggest DGDG is in high abundance? (pg.5 line 162). Do the authors have a comment on the importance of cardiolipin in the mitochondrial membrane for complex I activity, yet its absence here in the thylakoid membrane with the related NDH enzyme? Do they suspect a redundancy in lipid composition, a lack of importance for NDH, or some other reason?

Thank you for this interesting question.

We wrote “surprisingly” because, as you pointed out, DGDG is expected to be highly abundant in thylakoid membranes. However, this does not necessarily mean that DGDG associates preferentially with NDH. The review of Domonkos et al. (2008, 10.1016/j.plipres.2008.05.003) highlight structural studies that emphasize the importance of

PG and DGDG in photosynthetic complexes isolated from cyanobacteria, while PG plays a central role in photosynthetic complexes in vascular plants.

An analysis of protein structures within the photosynthetic electron transport chain from several published papers revealed that, for example, PSI shows a higher content of PG, followed by MGDG (Kobayashi et al., 2017, 10.3389/fpls.2017.01991). However, there are very few structures of NDH from vascular plants, including ours, and those from barley and *A. thaliana*, where the resolution is too low to differentiate lipids, which makes it difficult to distinguish a consistent pattern across models from different species. Interestingly, no DGDG density was observed in the published structures of plant NDH (Su et al. (2022) and Shen et al. (2021)), suggesting that DGDG may not bind to plant NDH.

To understand the role of different lipids within NDH, more structural studies of the complex from other photosynthetic organisms would be required. By comparing conserved and non-conserved lipid binding sites among different photosynthetic organisms, and by depleting specific lipids from the thylakoid membrane, it might be possible to gain insight into the universality and diversity of roles of individual lipid molecules in photosynthetic complexes, but this would go well beyond the scope of our present study.

Cardiolipin (CL) is essential for optimal activity of complex I in mitochondria (Fry and Green, 10.1016/S0021-9258(19)69888-1). It has been proposed that cardiolipin binds to the proton-pumping subunits of complex I, enabling global conformational changes that enhance the accessibility of the quinone substrate to the enzyme (Jussopow et al., 2019, 10.1126/sciadv.aav1850). CL is a unique phospholipid almost exclusively found in the inner mitochondrial membrane where it is biosynthesized. In contrast, thylakoid membranes are primarily composed of the galactolipids MGDG, DGDG and SQDG, and glycerollipid PG as pointed out in our manuscript.

In our supplementary table 1A, we show that although CL is not present in thylakoid membranes of *S. oleracea*, we observe lipid densities in positions occupied by CL in complex I from various species. These conserved positions across homologous complexes suggest that lipids at these specific sites are important. Other lipids may substitute for CL in plant chloroplasts to ensure the proper function of NDH.

7. “The PQ sits at the entrance of the cavity surrounded by several charged residues, interacting with Y259 of NdhA through its ketyl group (figure 7C).” (pg.6 line 196). I believe “ketyl” is incorrect here. It should carbonyl or ketone.

Thanks, changed to “carbonyl group” .

8. “Although we cannot rule out t-PCC α M (4-trans-(4-trans-propylcyclohexyl)-cyclohexyl α -maltoside, the detergent used for solubilising and purifying the supercomplex), the density in the PQ pocket looks too long and bulky for a detergent molecule, and PQ fits it almost

perfectly (figure 7C).” (pg.6 line 199) Outside of visual speculation, was the detergent docked into the region and compared to the PQ for best fit? Map–model correlation values?

Yes, we did. To evaluate the fit of PQ and detergent in the pocket, we used Phenix CryoEM Validation to run a "minimal model" comprising NdhA, NdhH, and NdhK (the three subunits forming the PQ binding pocket) bound either to PQ or the t-PCC α M molecule along with the composite map. This map was also reduced to the area of interest using the "volume zone" command in ChimeraX. The map-model mean correlation coefficient (CC) for the ligands was very similar, with values of 0.66 for PQ and 0.65 for the detergent molecule. According to the Phenix manual, "A CC value of 0.7 or better usually indicates that the ligand is at least partially correct. A value between 0.6 and 0.7 may not be entirely wrong, but should be treated with suspicion.". Given these similar results, we evaluated the density visually and referred to the literature. PQ is known to be the lipid-soluble electron donor of NDHI (Friedrich et al., 1995, 10.1016/0014-5793(95)00548-N; Endo et al., 1997, 10.1093/oxfordjournals.pcp.a029115; Corneille et al., 1998, 10.1016/S0005-2728(97)00074-1; Strand et al., 2017, 10.1074/jbc.M116.770792). Previous biophysical studies on *E. coli* complex I (Verkhovsky et al., 2012, 10.1016/j.bbabi.2012.04.013) and structural studies on both the NDH cyanobacterial isoform (Pan et al., 2020, 10.1038/s41467-020-14456-0) and complex I (Parey et al., 2019, 10.1126/sciadv.aax9484; Klusch et al., 2021, 10.1093/plcell/koab092) have shown that endogenous quinol molecules can be trapped within the chamber upon protein purification. Interestingly, in those structures, the trapped ubiquinone molecule was also visualized in the shallow site (Parey et al., 2019; Klusch et al., 2021). The observation of an elongated cryoEM density suggests the presence of a PQ molecule. PQ and its analogues are crucial compounds for key electron transfer reactions in photo- and oxidative-phosphorylation (Kaurola et al., 2016, 10.1016/j.bbamem.2016.06.016). The dynamic nature of PQ is essential for its function in photosynthesis, and this characteristic is reflected in the cryoEM density, where the extremities are better visualized at lower thresholds. Given these reasons, we hypothesize that the density corresponds to a molecule of PQ.

9. Serine is misspelled “serin” (pg.8 line 240).

Thanks, corrected.

10. “A t-PCC α M detergent and one MGDG (MGDG 1016/F, figure 8D) contribute to the formation of the funnel by blocking the lateral exit, interacting with NdhB, NdhE, the transversal helix of NdhF and PnsB1 (figure 6D, supplementary figure 7A, C).” (pg.8 line 257)—given the inclusion at this site of the detergent used in the purification, how trustworthy is funnel region definition in relation to the non-solubilised complex?

The funnel is well-supported. Indeed, the input channel at the interface between NdhB and NdhD (i.e., the NdhB-D channel) closely resembles that observed and described in *E. coli* complex I (Efremov and Sazanov, 2011, 10.1038/nature10330). We assume that subunit PnsB1, which is not present in *E. coli*, contributes to the funnel entry. Furthermore,

comparing our model with models of complex I revealed that both detergent and lipid molecules have been observed in a position similar to that of the t-PCC α M detergent molecule (see figure 3 below and supplementary table 1A). In the *T. elongatus* complex I, a molecule of the detergent digitonin was modeled (Zhang et al., 2020, 10.1038/s41467-020-14732-z). Similarly, in complex I from pig, a molecule of phosphatidylethanolamine was modeled (Gu et al., 2021, 10.1038/s41594-022-00722-w), and in the mouse complex I, cardiolipin was modeled (Agip et al., 2019, 10.1038/s41594-018-0073-1). Therefore, we it is safe to assume that the detergent molecule we observe substitutes for a lipid in this position before the complex was solubilized.

Figure 3: Superposition of different models of complex I with *S. oleracea* NDH reveals the presence of both detergent and lipid molecules in a position similar to t-PCC α M YAG 1020/F. Molecules are color coded by protein of origin.

Discussion:

11. “Our findings support previous studies indicating that particles with a 1:1 NDH to PSI-LHCI ratio predominantly preserve PSI-LHCI-2” (pg.10 line 307). In relation to my earlier point with this nomenclature, please specify here that “PSI-LHCI-2” is in reference to the position of PSI relative to NDH.

Thanks, added (now line 325).

12. Given the point of contact of the PSI to the NDH is predominantly from Lhca6 to NdhF, do the authors have a rationalisation for why this limited interaction of the PSI at this site is stabilising for the whole of the NDH, where pretty much all of subunit interactions are independent of PSI? For example, the “PSI-LHCI-1” position (Fig. S3) has far more contact with NDH. The authors highlight the predominance of this PSI position in the 1:1 supercomplex in the literature. Can they suggest other possible reasons (outside of NDH stability) for why this interaction would occur? Has a role for 1:1 proximity to ensure efficient cycling through membrane complex distribution been suggested in the literature?

As shown in Supplementary Figure 6 (now called Extended Data Figure 9), model inspection of PSI-LHCI-1 from both *A. thaliana* and barley revealed limited interactions with NDH. Specifically, PSI-LHCI-1 interacts with NDH primarily through a few contacts between the Lhca-5 TMH 2-3 loop and PnsB1 on the stromal side. In contrast, Lhca-6 interacts with NdhF and PnsB5 at both the stromal and luminal levels.

As mentioned in the text, there are three potential reasons for supercomplex formation.

First, supercomplex formation is crucial for the stabilization of NDH under excessive light conditions (Peng et al. 2009, 10.1105/tpc.109.068791; Kato et al., 2021, 10.1038/s41467-021-24065-0). Peng et al. demonstrated that in the *Arabidopsis* lhca6 mutant grown under controlled light conditions NDH levels were lower. Immunoblot analysis showed that NDH content was reduced to 80% in immature leaves and to 60% in mature leaves compared with the wild type. Additionally, NDH activity was affected: while still detectable in immature leaves of lhca6 (albeit slightly lower than in the wild type), it was absent in mature leaves, as indicated by chlorophyll fluorescence measurements. BN-PAGE analysis revealed the absence of the NDH-PSI supercomplex in both immature and mature leaves of lhca6, while other protein complexes remained unaffected. Interestingly, the post illumination increase in chlorophyll fluorescence was identical in lhca5 and the wild type in both immature and mature leaves (Peng and Shikanai, 2011, 10.1104/pp.110.171264). These data suggest that in immature leaves the formation of the NDH-PSI supercomplex is not connected directly to NDH activity, although in mature leaves its formation stabilizes the chloroplast NDH complex, in particular under high-light conditions (Peng and Shikanai, 2011). In contrast, NDH-mediated Fd-dependent PQ reduction was also detected in mature leaves of lhca5 lhca6 mutant in vitro (Peng and Shikanai, 2011, 10.1104/pp.110.171264). These contrasting results highlight the importance of the supercomplex formation between NDH and PSI-LHCI-2 (which contains Lhca-6), especially in mature leaves, in maintaining NDH stability and suggest its potential role for an efficient operation of NDH-dependent PSI cyclic electron transport in vivo throughout the leaf's lifespan. However further experiments are required to clarify if supercomplex formation is directly linked to the activity of NDH-CET.

Second, it could be an energy-saving process in mature leaves. Supercomplex formation seems to be connected to leaf age and is an energy-saving mechanism for stabilizing NDH and maintaining its functionality throughout the leaf's lifecycle, avoiding continuous NDH production. Kato et al. (2021, 10.1038/s41467-021-24065-0) found that the expression of assembly factors for NDH synthesis are preferentially active in immature leaves and not in mature ones. In angiosperms fully expanded leaves are used for long periods (from a month to a few years), and here NDH complex is stabilized via Lhca-6-dependent supercomplex formation rather than continuously producing it in mature leaves.

Third, the amount of PSI-LHCIs bound to NDH may be affected by environmental conditions. Proteomic comparisons of field-grown and lab-grown plants indicate changes in light-harvesting protein abundance, suggesting different light-harvesting strategies in natural conditions compared to lab conditions (Mishra et al., 2012, 10.1186/1471-2229-12-6; Flannery et al, 2021, 10.1002/pld3.355). Field conditions are rarely optimal and are subject to large fluctuations in light and temperature, lowering photosynthetic efficiency and causing photo-oxidative stress. Therefore, plants grown in natural conditions rely more on regulatory mechanisms to protect and repair photosynthetic machinery. The literature shows contrasting results regarding PSI-LHCI formation in different conditions. Flannery et al. found a 25%

reduction in PSI in field-grown plants compared to lab-grown plants, with a 40% increase in Lhca-5 and no change in Lhca-6. Conversely, Mishra et al. (2012, 10.1186/1471-2229-12-6) reported a significant reduction in Lhca-5 in field-grown plants.

In the liverwort *Marchantia polymorpha*, which lacks the genes coding for Lhca-5 and Lhca-6, the PSI–NDH supercomplex was not detected (Ueda et al., 2012, 10.1111/j.1365-313X.2012.05115.x). This suggests that these subunits were acquired during the evolution of land plants, possibly to enable the efficient operation of an NDH-dependent CEF pathway.

However, the activity and physiological significance of the NDH-PSI supercomplex in land plants remains unclear. Further experiments are necessary to precisely correlate NDH to PSI-LHCI ratios with variables such as temperature, light intensity, and plant maturity. Isolating the supercomplex from plants of the same species grown under both field and laboratory conditions, and harvested at different life stages, would help elucidate the roles of environmental factors and leaf age in supercomplex formation.

In the cyanobacterium *Synechocystis 6803* the formation of the NDH-1L-PSI supercomplex is directly linked to the activity of NDH-CET (Gao et al., 2016, 10.1104/pp.16.00585). In this organism, the deletion of *ndhL* impairs NDH-CET activity (Mi et al., 1992) and abolishes the NDH-1L-PSI supercomplex but does not affect the assembly of NDH-1L and NDH-1M complexes, even under high-light conditions. Gao et al. have shown that in the cyanobacterial mutant lacking CpcG2, a linker protein for PSI-specific antenna, NDH-1-dependent cyclic electron transport was impaired, highlighting the importance of CpcG2 in supercomplex formation. Deletion of CpcG2 destabilized NDH-1L and its degradation product NDH-1M, leading to a decrease in functional PSI centers and emphasizing the role of CpcG2 in maintaining the supercomplex integrity for efficient electron transport.

Model:

13. I noticed in the model that the coordination geometry of the FB and N6b clusters to some of their respective Cys ligands (Cys17 and Cys110) is a little off. As these are not the focus of the paper, I think it is okay, but be cautious in the future and consider imposing stronger restraints in regions of lower resolution that have FeS clusters.

We have refined the model with the latest release of PHENIX to improve our model.

Figures:

14. Figure 8. “(D) Close-up of the constriction in channel NdhB-D.”—please check. Should this be (G)? Add reference in the legend at G to the asterisk in this panel also.

Thanks, corrected.

15. Figure 8 A, B, E and G all show grey surface as the detected channel; however, D and F have the sliced view of the detected cavities (is this correct? —please add to legend if so). Please reconsider the colour choice/inclusion for the latter for D and F as it tends to obscure the information rather than help. Consider the simple model view or VDW radii or model surface representation in trying to visualise the cavity in the protein as the cavity surfaces are already represented in A and E.

We have re-drawn the figure.

16. Figures S10, 12 and 14 are referenced in the text before S9 (and S11 and S13). Reorder figures to reflect appearance in the text.

Figures reordered.

17. Supplementary figure 6 is referenced for the first time in the Discussion. This should be renumbered to follow the text citations (after current Fig S10, S12 and S14).

See above.

18. For supplementary figure 6, please state which cryoEM map is displayed in F (this work or EMBD 32477?).

Text revised as follows: “Extended Data Fig. 9: PnsB1 could mediate the interaction with the second PSI in *S. oleracea*. a, Superimposition of subunit PnsB1 from *A. thaliana* and *H. vulgare* to PnsB1 of *S. oleracea*. The structures are highly similar (RMSD compared to spinach: 0.86 for barley and 0.9 for Arabidopsis). b, Top view of the cryoEM map from *A. thaliana* (EMBD 32477). Models of PnsB1 and Lhca-5 from each species are fitted relatively to *S. oleracea* PnsB1...”

19. Supplementary figures 17 and 18 image labels read “wast” instead of “waste”.

Thanks, corrected

20. Supplementary figure 7 and 8 (and others): please add to the legend number and letters above the lipids/detergents indicated refer to the model chain IDs (I am assuming). This identifier nomenclature seems to be used throughout the manuscript so it may be worth adding a sentence to this effect in the main text somewhere.

Thank you, modified as follows: “Extended Data Fig. 6: NDH-PSI-LHCI-2 supercomplex contains at least 46 well-defined lipids and two detergent molecules. Overview of observed lipid densities in (a) NDH and (b) PSI-LHCI-2. Lipid models are fitted into the respective cryoEM density (contour level in the range of 0.2-0.4) that is colour coded by lipid identity: PG, salmon; MGDG, violet; SQDG, pink; DGDG, green. c, CryoEM density of the t-PCC(M

detergent (three letter code: YAG) in turquoise. Each lipid is identified by its residue number, followed by a backslash and the chain identifier. “

21. Supplementary figure 9: please add the threshold value for the map for the displayed map features and state which map was used (e.g., composite). Were there no map features in the PQ cavity (outside of the proposed PQ density)?

The unresolved features in Extended Data Figure 8, highlighted in cyan, were selected using the “vop subtract map othermap” command in ChimeraX. As “map” we used the map of voids, channels, and depressions within NDH generated with HOLLOW software tool³⁵ and as “other map” we employed various maps generated during data processing. The term “various” indicates that the different density maps originated from subsets with a different number of 3D refined particles that were generated during single particle data analysis. This approach was employed to minimize bias in identifying these map features that were consistently present. These features were selected and are visualized at a contour level of ~0.2. All maps were created after local refinement on the transmembrane arm of NDH, which is why we found no density in the PQ cavity.

22. Supplementary figure 10 legend sentence repeated: “. An * (asterisk) indicates positions which have a single, fully conserved residue. An asterisk indicates positions of single, fully conserved residues.”

Corrected

23. Supplementary figures 11–16 do not have the figure organism name “*S. oleracea*” in italics.

Corrected

Tables:

24. Supplementary table 2 is not referenced in the main text. Please add to a suitable location or consider removing it if not needed.

Removed

25. Please reference supplementary table 3 in the methods near PHENIX/Molprobitry.

Added. Since table 2 has been removed, table 3 is now table 2.

Methods:

26. “...in electron counting mode at a calibrated pixel size of 0.837 Å” (pg.12 line 386)—was

this a facility calibration or one done with your own references? If the latter please provide additional details i.e., what reference model.

We have modified the text as follows: “Micrographs were recorded in a Titan Krios G2 microscope operated at 300 kV (Thermo Fisher/FEI) on a K3 direct electron detector in electron counting mode at a calibrated pixel size of 0.837 Å. Calibration was done by the Central Electron Microscopy Facility of the Max Planck Institute of Biophysics using apoferritin, with an x-ray model as reference.”

27. “55.704 dose fractionated movies” (pg.12 line 386)—incorrect full stop placement: “55,704”

Corrected

28. “Details are shown in supplementary figures 17 and 18.” (pg.12 line 389). This references the purification supplementary figures in the cryoEM data collection section. I believe this may be incorrect placement. S19 instead, perhaps?

Corrected

29. Please add a reference for cryoSPARC (10.1038/nmeth.4169) outside of the specific

Reference added.

30. The Relion reference only refers to the Bayesian polishing improvement. Consider adding the reference to Relion-3 (as suggested was used in the reporting summary), which contains advancements such as CTF refinements (that were also carried out by the authors) (10.7554/eLife.42166).

Added.

31. Please provide information about where the real-space mask came from, Auto-generated? For the focus maps, please provide details for how these were made (molmap/volumes, extensions/dilations, soft edges). Consider adding the mask outlines to supplementary figure 2.

Thank you, we added this part to the Material and methods, paragraph “Image processing for single-particle cryo-EM” (page 14):

“Real-space masks were automatically generated by cryoSPARC. For Local Refinement, we created a mask base starting from molecular models using the molmap command in ChimeraX. The mask base was then converted into a mask using Volume Tools in cryoSPARC. In this process, we set the threshold values chosen in ChimeraX (which varied

from mask to mask), set the dilation radius to 0 pixels, and the soft padding width to 8 pixels. All other parameters were left at their default settings”.

We have modified supplementary figure 2 (now Extended Data Figure 4) adding clear masks used for the local refinement jobs.

32. Please verify you mean to cite UCSF Chimera and not UCSF ChimeraX (the reporting summary states ChimeraX v1.2). The latter is used for ISOLDE, so it should be included as a citation anyway. Please use the citation most in line with the version of ChimeraX used (I think it is probably 10.1002/pro.3235). Remove UCSF Chimera reference if this was not used (not listed in the reporting summary).

Revised as suggested.

33. The reporting summary suggested the version of PHENIX used was much later than the reference used for the citation; consider updating the reference to something more current (doi:10.1107/S2059798319011471).

Revised as suggested.

34. Please provide the probe size used in the HOLLOW software to generate the tunnels. This will help with comparisons to other software.

Revised as follows: “To identify tunnels, cavities, and voids in the NDH complex, we utilized the software tool HOLLOW³² with default values (i.e. probe radius of the dummy atoms set to 1.4 Å and the grid-spacing to 0.5 Å).”

35. You started with an initial blob pick of ~17 million particles, of these if got narrowed down to ~4.5 million particles. Subsequently, these were refined to 45k and 168k particles, before being grouped to 107,752, cleaned again to 88,276 before a final classification to 38,385 particles. The final particle number is an incredibly small percentage of the starting material. The authors should provide more information on the types of classes that are dismissed. Are they all junk particles? Are they of varying compositions? Fig. S19 should be expanded to include information about examples of dismissed volumes at the E and F stages, which would be more informative than repeating the same final volume from each Topaz picking pipeline. There is space on the figure for both. In the CN-PAGE gel you demonstrate presence of isolated PSI and CF1Fo. Were these detected in the cryoEM analysis also?

We have revised Supplementary Figure 19, now renumbered as Supplementary Figure 10 following the figure numbers check. In panel e, you can see all the components into which particles were divided during the 3DVA analysis and subsequently displayed. Notably, components 0 to 4 (i.e., comp 0-4) do not show a well-defined peripheral arm, whereas components 5 to 9 exhibit a clear density map for the SubA and E domains.

Regarding the presence of isolated PSI and CF1Fo in the cryoEM analysis, the answer is yes. CF1Fo was clearly observed during the 2D classification (see figure 4 below) and these particles were subsequently excluded. Isolated PSI particles were not distinguishable in the 2D classification, likely because they resembled those bound to NDH. We believe that further rounds of heterogeneous refinement successfully removed these particles.

Figure 4: Excluded 2D classes showing CF1Fo.

Datasets merging:

1. Check for duplicated particles;
2. 2D classification;
3. 3D variability analysis (3DVA) focused on whole complex
4. 3D variability display (cluster mode, 3 clusters)
5. Re-extraction particles from cluster 0: 107,752 particles, 0.837 Å pixel size
6. NU-refinement **107,752 particles**, 3.48 Å (FSC =0.143) (a and b):

Particles exported to Relion:

- | | |
|-------------------|-------------------|
| 24,554 exposures: | 30,876 exposures: |
| 1. MotionCor2; | 1. MotionCor2; |
| 2. CTFFind 4; | 2. CTFFind 4; |

3. Extraction using imported coordinates (**107,752 particles**, 0.837 Å pixel size);
4. 3D refinement;
5. Ctf refinement;
6. Bayesian Polishing;
7. Ctf refinement (3.42 Å (FSC =0.143));

Particles re-imported in cryoSPARC:

1. Heterogeneous refinement;
2. NU-refinement (**88,276 particles**, 3.22 Å (FSC =0.143));
3. 3DVA (mask on peripheral arm (c), 3 modes);
4. 3D variability display (intermediates mode, 10 frames, window 0, output particle subset ON, output particle component 2 (d, e));
5. NU-refinement (**38,385 particles** from frames 5-9 of component 2 (d, in green and e), 3.34 Å (FSC =0.143), Extended data Fig. 4a-c);
6. Local refinements (results shown in Extended Data Fig. 2).

Supplementary Fig. 10: CryoEM processing pipeline and final map for NDH-PSI-LHCI-2. Selected particles from the two datasets, picked using different settings in Topaz (see Supplementary Fig. 7e, f), were merged. The workflow leading to the final map is

detailed. **a**, and **b**, Representative cryo-EM map and corresponding FSC curve of NDH-PSI-LHCI-2 before particles selection via 3D variability analysis (3DVA), showing a poorly resolved peripheral arm. **c**, The mask used for 3DVA around the peripheral arm of NDH, is shown in yellow. **d** and **e**, Particle subsets were generated using 3D variability display on component 2 (a total of three modes was solved in 3DVA), utilizing the "intermediates" mode with the "rolling window" set to 0. This setting ensured an even distribution of particles along the chosen component (**d**). Volumes generated from each subset were analyzed in ChimeraX. Only those showing a well refined peripheral arm were chosen for further 3D refinement. Maps are shown at a threshold level of 1. From the resulting subsets, subsets five to nine (**d**, highlighted in green and **e**) were selected for further refinement. The 3D reconstruction of the merged particle clusters five to nine resulted in a map with a well resolved peripheral arm of NDH (see Extended Data Fig. 3a-c).

36. "3D-refined particles with a real-space mask around the peripheral arm were used as inputs for 3D variability analysis (3DVA) in order to separate particles that showed a well-defined peripheral arm from those where it was damaged." (pg.13 line 403). Please better define "damaged". Low-resolution regions (local resolution maps)? Missing subunits?

Thank you for this comment. By "damaged" we mean particles that yielded a map in which the peripheral arm of NDH was fuzzy and clearly incomplete. To clarify, we updated Supplementry Fig. 10 as shown in the previous answer to question #35.

37. "Subsequently, the models were docked into the various maps using UCSF Chimera" (pg.13 line 421). Please specify what the "various maps" were. Each respective focus map used to build that specific region?

Revised as follows: "Subsequently, the models were initially docked into the various locally refined maps using UCSF ChimeraX⁶⁵. Where necessary, amino acid residues of specific subunits were modified according to the spinach sequence in Coot ⁶⁷."

38. "Initial model fitting was conducted using ISOLDE⁶⁷, and iterative refinement was performed with Coot⁶⁶ in conjunction with PHENIX". (pg.13 line 423) Please specify which map was used for the final refinement i.e., the consensus cryoSPARC map or the composite map.

Revised: "Initial model fitting was conducted using ISOLDE⁶⁸, and iterative refinement was performed with Coot ⁶⁷ in conjunction with PHENIX⁶⁹. Final model fitting was performed using the composite map (EMDB: 19210) with Coot ⁶⁷ in conjunction with PHENIX⁶⁹. The quality of the models was assessed with Molprobit⁷⁰ (Supplementary table 2)."

39. Please add the Rama-Z scores from PHENIX validation to Supplementary Table 3.

Scores added.

References:

Please be more careful about referencing the appropriate resource for cryoEM/model building software that you used (examples noted above).

Authors should review the references in the bibliography and take more care with the reviewing the formatting. Examples of issues are non-italicised organisms and unwanted characters such as `it` and `*` in titles (refs 67, 68, 13, 17, 18, etc.)

Thank you, this has been checked.

General:

- Is there any functional data (here or in the literature) to support that the sample in the structural analysis can perform its predicted function (with respect to the absent subunits)?

It was not possible to perform functional test with our Sample. Direct genetic manipulation of plants is difficult at best. Moreover, most large complexes of the thylakoid membrane are of similar size and have similar isoelectric points which makes it difficult to isolate NDH-1-PS1 complex by Ion-Exchange and Size Exclusion Chromatography at a concentration or purity that would be required for reliable biochemical analysis – especially considering the relatively low abundance of NDH-1-PS1 (Suppl. Fig. 10 and 11). A large part of our protein purification was performed at the stage of image processing *in silico* by 2D and 3D image classification, which is the main reason why only less than 40.000 particles remained in the final data set.

- Are there any mutants with knock outs (in related model organisms) of the absent subunits in this structure and how do they behave? This may inform whether the missing subunits are necessary for catalysis.

Yamamoto et al., 2011a and Fan et al., 2015 demonstrated that a knock-out or knock-down of either NdhU, NdhS, NdhT or NdhV effectively abolishes the ETR in a *pgr5* mutant background. Since both NdhS and NdhV can dissociate from the NDH-1 complex in the presence of detergent, it is not unexpected that they are lost during purification. Similar results for NdhV have been observed in Cyanobacteria (Gao et al., 2016, 10.1104/pp.15.01430).

While NdhU is not required for the accumulation of other NDH subunits, it is essential for NDH activity. Conversely, the accumulation of NdhU depends on NdhT and SubA,

which contradicts our findings (Yamamoto et al. 2011, 10.1105/tpc.110.080291). Furthermore, biochemical assays indicated that NdhS is essential for efficient Fd-dependent plastoquinone reduction *in vitro*. Yamamoto and Shikanai (2013, 10.1074/jbc.M113.511584) produced *A. thaliana* plants with point mutations in the positively charged pocket of the SH3-like domain of NdhS. Nine charged residues in this domain are highly conserved among plants. Systematic alteration of these sites, particularly Arg 139 to neutral glutamine, revealed a decrease in NDH activity *in vivo*. Further substitution of Arg 193 with negatively charged aspartate or glutamate or hydrophobic alanine significantly reduced the efficiency of ferredoxin-dependent plastoquinone reduction by NDH in ruptured chloroplasts. Similar results were obtained from *in vivo* analyses of NDH activity and electron transport. However, the mechanism of interaction between NdhS and ferredoxin in land plants remains unclear, and structural data are needed to elucidate this process.

Fan et al. (2015, 10.1111/tpj.12807) demonstrated that in the *Arabidopsis ndhv* mutant, NDH SubA and SubE degrade more rapidly than in the wild type under high-light treatment, resulting in impaired NDH activity.

CryoEM structures of cyanobacterial NDH from *C. thermophilus* binding ferredoxin (Fd) (Zhang et al., 2020, 10.1038/s41467-020-14732-z) revealed that Fd interacts directly with NDH at the previously proposed O-site (Laughlin et al., 2019, 10.1038/s41586-019-0921-0), which includes NdhO, NdhI, NdhH, and NdhV. Interestingly, NdhV and Fd were added to the final preparation for cryoEM, but no direct interaction with NdhS was observed. Pan et al. (2020, 10.1038/s41467-020-14456-0) resolved the structure of cyanobacterial NDH from *T. elongatus* in the presence of Fd and found that the Fd binding pocket is mainly formed by NdhI and NdhH. Both NdhS and NdhV were present but did not directly interact with Fd. Interestingly, the J and J-like proteins NdhT and NdhU are not part of the cyanobacterial complexes.

- NdhT and NdhU look very structurally similar. Do authors predict redundancy in these subunits at the modelled position? A possible role as an assembly factor? A brief comment in the Discussion on the predicted differences between this structure and the native structure with these included 'missing' subunits may be useful.

Based on the above results, we conclude that NdhU in NDH from land plants may optimize the structure of the peripheral arm or the interaction with the connecting subunits of this domain. NdhU binding may trigger the binding of NdhT, NdhS, and NdhV, ultimately resulting in Fd binding, which at present is however purely speculative. Purifying the complex in the presence of the missing subunits or chemical crosslinking might provide structural insights into a more complete supercomplex, but both are well outside the scope of our present study.

- Why the choice of this detergent over others?

In one of our earlier publications (DOI: [10.1126/science.aat4318](https://doi.org/10.1126/science.aat4318)), we conducted a comprehensive detergent trial which found that tPCCaM was preferable over nanodiscs for solubilizing large intact and flexible thylakoid membrane complexes, whereas DDM or digitonin were less suitable. For the chloroplast ATP synthase, we demonstrated by negative stain-EM that all expected protein subunits/subcomplexes remained intact with no indication of altered abundance within the complex, which did occur in DDM or SMALPS. Negative-stain EM of initial NDH-1-PSI purification trails confirmed our earlier positive results for tPCCaM.

Remaining reviewer comments

Our guidance:

Please revise the text to address the remaining concerns of the reviewers (pasted again below) and submit a separate point-by-point response. Please ensure that all reviewer comments are addressed.

Reviewer #1 (Remarks to the Author):

Second round review of “CryoEM structure of NDH-PSI-LHCI supercomplex from *Spinacia oleracea*”.

The authors took the effort to address all issues and questions raised by us. Their corrections and explanations, both in the Manuscript as well as in their “Response to reviewers” are fully satisfactory. Thank you also for the non-essential clarifications.

We recommend the work for publication.

TYPOS:

- In the main text “Supplementary figure(s)” is spelled with both capital and small “S”.

Reviewed by Ricardo Righetto and Wojciech Wietrzynski

Reviewer #2 (Remarks to the Author):

The authors have adequately addressed all of the reviewers comments and no further revisions are needed.

Reviewer #3 (Remarks to the Author):

The authors have addressed all aspects that were brought up in the review process well.

Minor comments: Supplementary figure 1 has a repeated 'from' in its title and Supplementary Table 2 is lacking a space in the 'Datacollection and processing' column header.

Answers to reviewers:

Answer to reviewer #1:

Thank you. We made the corrections.

Answer to reviewer #2:

Thank you.

Answer to reviewer #3:

Thank you. We made the corrections.